

# Non-polar organic compounds in aerosols in a typical city of Eastern China: Size distribution, gas-particle partitioning and tracer for PM$_{2.5}$ source apportionment

Deming Han[1], Qingyan Fu[2], Song Gao[2], Hao Xu[1], Shan Liang[1], Pengfei Cheng[3], Xiaojia Chen[1], Yong Zhou[1], Jinping Cheng[1]

[1] School of Environmental Science and Engineering, Shanghai Jiao Tong University, Shanghai 200240, China

[2] Shanghai Environmental Monitor Center, Shanghai 200235, China

[3] School of Chemical and Environmental Engineering, Jiujiang University, Jiujiang 332005, Jiangxi, China

*Correspondence to*: Jinping Cheng (jpcheng@sjtu.edu.cn)

**Abstract.** Aerosol-associated non-polar organic compounds (NPOCs), including 15 polycyclic aromatic hydrocarbons (PAHs), 30 n–alkanes, 2 iso–alkanes, 5 hopanes and 5 steranes, were identified and quantified in PM$_{2.5}$ samples using thermal desorption–gas chromatography/mass spectrometry (TD-GC/MS) method. The samples were collected in a typical city of Eastern China. The total concentrations of NPOCs were 31.7–388.7 ng m$^{-3}$, and n–alkanes were the most abundant species (67.2%). The heavy molecular weight PAHs (4- and 5-ring) contributed 67.88% of the total PAHs, and the middle chain length n–alkanes (C$_{25}$–C$_{34}$) were the most abundant in n-alkanes. PAHs and n-alkanes were majorly distributed in 0.56–1.00 μm fraction. ∑(hopanes+steranes) were associated with the 0.32–1.00 μm fraction. Analysis showed that 83.0% of NPOCs were originated from anthropogenic sources, especially pyrogenic sources such as fossil fuel combustion and biomass burning. The ratio–ratio plots indicated that NPOCs in local area were affected by photochemical degradation and emissions from mixed sources. Gas-particle partitioning model showed that the particle-phase fraction (φ) of light molecular weight NPOCs ranged from 2.4% to 62.5%, while that of heavy NPOCs accounted for more than 90.0%. The data based on single particle phase and the data based on gas-particles phases incorporated with other PM$_{2.5}$ compounds were used as input data for positive matrix factorization (PMF) model, respectively. Eight factors were extracted for both cases: secondary aerosol formation, vehicle exhaust, industrial emission, coal combustion, biomass burning, ship emission, dust and light NPOCs. This study provides new information on the profiles of PM$_{2.5}$-associated NPOCs, size-specific distributions, photodegradation and their gas-particle partitioning. This will help us accurately identify the potential sources of aerosols and then asses the contributions from each source.



## 1. Introduction

In recent years, severe atmospheric pollution characterized by haze has been occurring in developing countries (Yadav et al., 2013;Wang et al., 2015), affecting visibility, optical radiation and human health (Shen et al., 2015;Sulong et al., 2017). China has experienced numerous severe and long-lasting haze episodes since winter in 2013, which has affected over 600 million local residents and covered a quarter of the country's land area (Huang et al., 2014;Hao and Liu, 2015). In essence, haze episode is caused by the distribution of particle matters with different sizes in atmosphere, leading to decrease in visibility (Xie et al., 2017). Carbonaceous aerosols contain a large amount of particle matters, accounting for 30–50% of $PM_{2.5}$ mass concentrations (Yadav et al., 2013;Wang et al., 2016). Carbonaceous aerosols have significant influence on environmental and physical processes, including dry/wet deposition, cloud condensation nucleation, and heterogeneous reactions (Li et al., 2017;Feng et al., 2006).

Since carbonaceous aerosols can affect ambient environment significantly, it is crucial to investigate aerosol-associated organic compounds. Because non-polar organic compounds (NPOCs) can provide specific information on the identification of aerosol sources (Rajput and Sarin, 2014;Wang et al., 2016), they are now of special interest to researchers. n-Alkanes, polycyclic aromatic hydrocarbons (PAHs), hopanes and steranes are four typical NPOC species, and are frequently used to apportion sources of ambient particulate matters (Xu et al., 2013;Zhao et al., 2016). n-Alkanes are emitted from natural and anthropogenic activities, including particulate abrasion products from leaf surfaces of vegetation and fossil fuel combustion. Notably, fossil fuel combustion is characterized by the release of $C_{22}$–$C_{25}$ n-alkanes, while particulate abrasion products from leaf surfaces is characterized by the predominance of >$C_{29}$ odd n-alkanes (Yadav et al., 2013). PAHs are mainly emitted from anthropogenic activities, including biomass burning, coal combustion, fossil fuel combustion and industrial processes (Ma et al., 2011;Zhang et al., 2015). Hopanes and steranes are from unburned fossil fuels and lubricant oils, and they are often found in vehicle exhausts, ship emissions and coal combustion emissions.

PAHs are typical semi-volatile organic compounds, which can partition between gas and particle phases in ambient atmosphere (Ma et al., 2011;He and Balasubramanian, 2009). The gas-particle partitioning behavior of semi-volatile organic compounds can be used in their removal process, which is performed through wet/dry deposition, long-range transport and reaction with atmospheric oxidant. Recently, research has shown that n-alkanes, hopanes and steranes are also semi-volatile and subject to gas-particle partitioning (Xie et al., 2013;Xie et al. 2014;Wang et al., 2016). Furthermore, lighter, more volatile compounds show larger partitioning tendency. Additionally, another crucial factor affecting particle-bound concentrations of NPOCs is their aerodynamic diameter (Wang et al., 2009). The size-specific



distributions of compounds are dependent on their physical-chemical properties, gas-particle partitioning and
photodegradation (Okonski et al., 2014;Chen et al., 2016b). Thus, characterization of the size-specific distribution of
NPOCs is crucial for understanding their formation, assessing their possible environmental fate and offering proper
management (Kleeman et al., 2008;Wang et al., 2009). Although several studies (Hien et al., 2007;Wang et al., 2009)
have focused on the size distribution of PAHs, much less attention has been given to other NPOC species.
NPOCs are typically assumed to be stable and nonreactive (Feng et al., 2006;Ma et al., 2011). However, recent research
(Xie et al., 2013;Wang et al., 2016) has shown that NPOCs can be oxidized by $\cdot$OH radicals, $RO_2\cdot$radicals and $O_3$ over
atmospherically relevant time scales. PAHs, n-alkanes, hopanes and steranes can undergo photochemical oxidation,
increasing the production of secondary organic aerosol (Robinson et al., 2006;May et al., 2012). For example, PAHs can
react with $\cdot$OH radicals, and through adding carbonyl groups to the carbon skeleton, the free ends of C-C scission
products remain tethered together, which prevents fragmentation and leads to more functional groups on the single
product (May et al., 2012). Hence, the low-volatility species which can condense into particle phase are formed, and
subsequently undergo oligomerization reactions following condensation.
To develop strategies for controlling atmospheric pollution caused by particulate matter, receptor-based models (e.g.,
positive matrix factorization, PMF) have been widely applied to quantitatively apportion sources of particulate matter
(Wang et al., 2009;Li et al., 2016;Huang et al., 2017). However, the output factors of receptor model are not necessarily
emission sources, because there exist some atmospheric processes like photodegradation or gas-particle partitioning.
Considering the gas-particle partitioning of NPOCs, Xie et al. (2013) adopted both gaseous and particulate NPOCs in
PMF model and successfully extracted seven factors. More recently, Wang et al. (2016) used data of NPOCs combined
with those of organic/elemental compounds (OC/EC), inorganic compounds and elemental compounds as input for PMF
model, and they found that total (gas+particle) bound concentrations enabled more reasonable source profiles than single
particle phase.
In this paper, we conducted a comprehensive study on $PM_{2.5}$-associated NPOCs in a typical city of Eastern China.
Specifically, we: (1) quantified the concentrations of NPOCs (n–alkanes, PAHs, hopanes and steranes) through thermal
desorption–gas chromatography/mass spectrometry (TD–GC/MS) method; (2) determined the size-specific distributions
of NPOCs from 0.01 to 18 μm; (3) analyzed the degradation of NPOCs; (4) explored the gas-particle partitioning of
NPOCs and assessed its effects on $PM_{2.5}$ source apportionment.




**2. Materials and methods**
**2.1 Sampling sites and sample collection**
Jiujiang city is located in 113°57'–116°53' E and 28°47'–30°06' N with elevation of 32 m in Jiangxi Province of Eastern
China. It is characterized by a subtropical monsoon climate. Jiujiang is the second largest city in Jiangxi Province, with
approximately 4.83 million resident population and over 700 thousand motor vehicles in 2015. Preliminary statistics
indicate that the gross industrial standard coal consumption in Jiujiang amounted to 7.80 million tons in 2015. In Jiujiang,
petrochemical industry, which can process approximately five million tons of crude oil per year, is located at the
northeast part of the city and in upwind direction. In addition, Mount Lu (elevation of 1474 m), located at the south of
Jiujiang, blocks the transport of air masses from Northern China Plain region to southern area, leading to the
accumulation of particulate matters in the city area, especially in winter seasons.
Five PM$_{2.5}$ sampling sites were selected for routine air quality measurements in Jiujiang city (Table 1), including Shihua
(SH), Xiyuan (XY), Shili (SL), Wuqierqi (WQ) and Jiujiangxian (JJ). High-volume air samplers (YH-5, Qingdao, China)
loaded with a quartz fiber filter were used. Prior to use, they were prebaked at 550 °C for 4 h to eliminate residual organic
compounds. PM$_{2.5}$ sampling at these five sites were performed synchronously for five continuous days per month from
Sep. to Dec. 2016, in addition to the extensive sampling period (Dec. 1–16).
**Table 1.** Detailed description of the six sampling sites in this study

| Site | Type | Surrounding area | Height | Main pollutants | Sampling time |
|------|------|------------------|--------|-----------------|---------------|
| SH | Residential area | 1.7 km to petrochemical industry; 600 m to traffic road | ~12 m | Petrochemical industry | Sep. 9-13; Oct. 11-15; Nov. 10-14; Dec. 1-16 |
| XY | Urban area | 500 m to traffic road; 1 km to wharf | ~23 m | Vehicle; ship | Sep. 9-13; Oct. 11-15; Nov. 10-14; Dec. 1-16 |
| SL | Urban center | 500 m to traffic road | ~20 m | vehicle | Sep. 9-13; Oct. 11-15; Nov. 10-14; Dec. 1-16 |
| WQ | Suburban area | 1.3 km to the city center | ~17 m | vehicle | Sep. 9-13; Oct. 11-15; Nov. 10-14; Dec. 1-16 |
| JJ | Industrial area | Located in industrial area; 500 m to highway | ~20 m | Industry emission | Sep. 9-13; Oct. 11-15; Nov. 10-14; Dec. 1-16 |
| EM | Urban area | 500 m to traffic road | ~20 m | vehicle | Dec. 1-16 |

The sampling site of size–specific aerosols was located at a five–story building of Jiujiang Environmental Monitor
Station (EM site). Airborne particle samples were collected for 23 h using a Nano–Micro–orifice Uniform Deposition
Impactor (MOUDI) sampler (Model 122R, MSP Cor, USA), at air flow rate of 30 L/min. Detailed instrument operation,
quality assurance and control method can be found in our previous work (Chen et al., 2016b). Briefly, this sampler can
collect particles within 13 size fractions: 0.01–0.018, 0.018–0.032, 0.032–0.056, 0.056–0.1, 0.1–0.18, 0.18–0.32,



0.32–0.56, 0.56–1.0, 1.0–1.8, 1.8–3.1, 3.1–6.2, 6.2–9.9, 9.9–18 μm. Prior to sampling, each filter tray was washed with
distilled water and ethanol. Two kinds of quartz fiber filters (diameters of 47 and 90 mm, respectively) were prebaked at
550 ºC for 4 h, wrapped in aluminum foil and then sealed in clean polyethylene bags. Leak and flow tests were conducted
according to manufacturer's instructions: leak test was done with a duration of 60 s with leak rate < 10 Pa/s at initial
pressure of 55±5 kPa. The mass concentrations of particle matters were determined by subtracting the filter weight before
and after sampling. A well calibrated digital balance within a precision of 0.01 mg (Sartorius SE2, Germany) was used.
The collected particle samples were stored at controlled temperature (–20 ºC) and relative humidity until analysis.

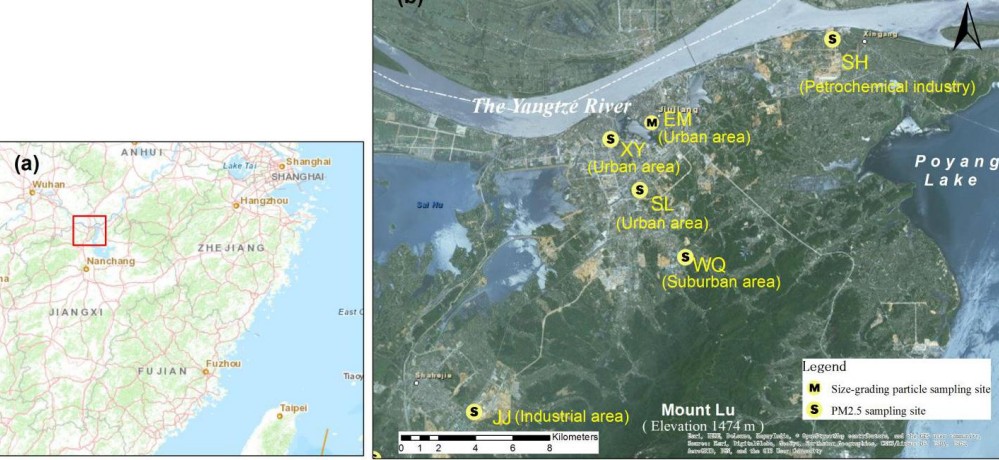


**Fig. 1.** Location of aerosol sampling sites in Jiujiang, Eastern China
**2.2 Analysis of aerosol samples using TD–GC/MS**
Fifty-seven NPOC species (Table S1) were identified by using an in-injection port thermal desorption union (TDU,
Shimadzu, Japan), coupled with gas chromatography/mass spectrometer (GC/MS, QP2010 Plus, Shimadzu, Japan).
Compared with traditional solvent extraction method, TD–GC/MS method (Ho and Yu, 2004;Ho et al., 2008) has
advantages such as solvent and sample filtration, labor saving, and less contamination from solvent impurities. A filter
aliquot (1 cm$^2$) from quartz fiber was cut into small pieces on a clean glass dish, and then they were inserted into the TD
tubes (CAMSCO, USA). Both sides of the samples were surrounded with pre-baked, silane-treated glass-wool plugs, to
enhance the cryofocusing of the analytes and prevent heavy and polar compounds from entering the GC column.
The sample processing time in TD tube was set to 45 min, and the TD tube was electronically cooled to −10 ºC. The
desorption and interface temperatures were set to 295 and 280 ºC, respectively. Helium (99.999%) was used as carrier
gas for the thermally desorbed organic compounds, with gas flow rate of 1.12 mL/min. GC was used under splitless



injection mode, and the initial oven temperature was set to 40 °C with an isothermal hold time of 5 min. Stepwise
programmed linear temperature ramping included 10 °C/min to 120 °C (held for 2 min), and then 20 °C/min to 300 °C
(kept for 20 min). A Rtx–5MS capillary column (Restek, USA, L × I.D. 30 m × 0.25 mm, df 0.25 μm) was used to
separate desorbed organic compounds. Mass range was m/z 50–500 and scanned at 0.5 s/scan. The ion was produced
from electronic impact ionization (EI) at 70 eV, and then was separated by high performance quadrupole mass filter.

**2.3 Determination of OC/EC and other constituents**

OC and EC were analyzed (a round punch of 0.538 cm$^2$) using the thermal–optical– transmittance (TOT) method
(NIOSH protocol, Desert Research Institute, USA). The instrument included a temperature- and atmosphere-controlled
oven and a laser of 680 nm wavelength to generate an operational EC/OC split. The instrument was heated stepwise from
start to 250 °C (60 s), 500 °C (60 s), 650 °C (60 s) and finally 850 °C (90 s) in the helium atmosphere for OC
volatilization, and from start to 550 °C (45 s), 650 °C (60 s), 750 °C (60 s) and finally 850 °C (80 s) in the helium
atmosphere containing 2% oxygen for EC oxidation.
Elemental compositions, including Na, K, Ca, Mg, P, Fe, Ti, Al, Pb, Cu and Zn, were determined by energy dispersive
X-ray fluorescence (ED-XRF) spectrometry (Epsilon 5, Netherlands). Water soluble inorganic ions, including cations
(Na$^+$, K$^+$, Mg$^{2+}$, Ca$^{2+}$, NH$_4^+$) and anions (Cl$^-$, SO$_4^{2-}$ and NO$_3^-$, NO$_2^-$), were detected by ion chromatography (IC, ISC-90,
Dionex, USA).

**2.4 Quality assurance and quality control**

Prior to sampling in each site, the five PM$_{2.5}$ samplers were calibrated by environmental monitor station. PM$_{2.5}$ samplers
were placed on the rooftop at EM site with distance between any two samplers <3 m, collecting for 12 h, then the added
mass was calculated. Field blanks were collected by keeping blank filters in the sampler for the same duration at
sampling site. Additionally, both transport and laboratory blank filters were analyzed, and all the data reported in this
study were corrected according to the results.
The internal standards of n–tetracosane d$_{50}$ (n–C$_{24}$D$_{50}$), naphthalene–d$_8$, acpnaehthene–d$_{10}$, phenanthrene–d$_{10}$, and
chrysene–d$_{12}$ were spiked into each sample. This was done to account for the loss of components from sample filters
associated with the instrument instability due to changes in laboratory environmental conditions. SRM 2260, SRM
2260A and SRM 1494 (NIST, USA) were used as calibration standards for PAHs, n–alkanes and hopanes/steranes,
respectively. Six–point calibration curves of NPOCs were constructed through adopting different calibration solutions,
namely 0, 0.05, 0.25, 0.50, 1.0 and 2.0 μg L$^{-1}$ for PAHs, hopanes and steranes, while 0, 0.10, 1.0, 5.0, 10.0 and 20.0 μg





L$^{-1}$ for n–alkanes, to the field blank filters and loaded into TD tubes. Ratios of the average peak area values 'A' for
represented samples 'S' to the corresponding internal standard 'IS', namely AS/AIS; and ratios of average concentration
'C' for represented samples to the corresponding internal standard (CS/CIS), were generated via using Shimadzu
software with the slope of the curve being the RRF (relative response factor). The calibration curves for most target
compounds were highly linear (r$^2$>0.99), demonstrating the consistency and reproducibility of this method. The method
detection limit (MDL) was determined at the 99% confidence level.
**2.5 Diagnostic parameters and isomeric ratios**
Different diagnostic parameters were adopted in this study to explore natural and anthropogenic contributions. The
parameters include carbon preference index (CPI), the carbon number of the most abundant n–alkane (C$_{max}$),
contributions from natural wax n–alkanes (WNA%) and petrogenic n–alkanes (PNA%), higher plant n–alkane average
chain length (ACL), and molecular diagnostic ratios (MDRs) for PAHs (Yadav et al., 2013;Zhao et al., 2016).
(1) CPI is defined as the ratio of the total concentration of odd n–alkanes to that of even n–alkanes (Eq. (1)). It reflects
the comparison between natural and anthropogenic contributions.
$$CPI = \frac{\sum_{i=11}^{39} C_i}{\sum_{j=12}^{40} C_j} \qquad (1)$$
where i and j represent odd and even carbon numbers, respectively, and C represent the concentrations of carbon
n–alkanes.
(2) WNA% is calculated as Eq. (2). Note that negative value of [C$_i$ – (C$_{i-1}$ + C$_{i+1}$)/2] was replaced by zero.
$$WNA\% = \frac{\sum WNA_{Cn}}{\sum NA_{Cn}} = \frac{\sum_{i=11}^{39} [C_i - (C_{i-1} + C_{i+1})/2]}{\sum_{n=11}^{40} C_n} \qquad (2)$$
(3) Aerosol-associated n–alkanes can originate from plant wax or petroleum combustion, and the percentage of
petrogenic n–alkanes (PNA%) can be calculated as:
$$PNA\% = 100\% - WNA\% \qquad (3)$$
(4) C$_{max}$ represents the carbon number of the most abundant n–alkane, and it is regarded as the most important indicator
of biogenic inputs. In general, C$_{max}$ = 31 indicates effects from leaf abrasion products, whereas C$_{max}$ = 29 implies effects
from road dust, as well as vehicle and industrial emissions.
(5) ACL can indicate emissions of n–alkanes from plants which are related to temperature and humidity. It is defined as
counted carbon atoms per carbon molecule, depending on the odd n–alkanes from higher plant. ACL can be estimated
through Eq. (4) as follows:




$$ACL = \frac{23 \cdot C_{23} + 25 \cdot C_{25} + \cdots + 39 \cdot C_{39}}{C_{23} + C_{25} + \cdots + C_{39}} \qquad (4)$$
(6) MDRs for PAHs source apportionment include ANT/PHE ratio, PYR/FLU ratio, IcdP/BghiP ratios (namely,
ANT/(ANT+PHE) ratio and FLU/(FLU+PYR) ratio), and IcdP/(IcdP+BghiP). If ANT/(ANT+PHE) < 0.1, petroleum
origins are suggested; if the ratio > 0.1 pyrogenic sources are indicated. If FLU/(FLU+PYR) < 0.4, petroleum sources are
suggested; if the ratio > 0.4, pyrogenic sources are indicated (Kuang et al., 2011;Chen et al., 2016a). Note that the ratio
ranging from 0.4 to 0.5 also suggests fuel combustion. If 0.2 < IcdP/(IcdP+BghiP) < 0.5, fuel combustion is suggested; if
the ratio > 0.5, grass, wood and coal combustion should have contributed to particulate matters(Chen et al., 2016a).
**2.6 Gas-particle partitioning model**
Gas-particle partitioning is an important mechanism that affects the fate and transport of NPOCs (Pankow, 1994;Kim et
al., 2011). To understand the partitioning behavior of NPOCs, we evaluated the distribution of NPOCs between gas and
particle phases in the atmosphere. The gas-particle coefficient $K_p$ (m$^3$ μg$^{-1}$) for each compound species was calculated
using the following equations:
$$K_p = \frac{F/PM}{A} = \frac{R \cdot T}{10^6 \cdot MW_{OM} \cdot \xi_{OM} \cdot P_L^o} \qquad (5)$$
$$P_L^o = P_{L,0}^o \cdot \exp\left[ \frac{\Delta H_0}{R} \left( \frac{1}{298} - \frac{1}{T} \right) \right] \qquad (6)$$
where F and A represent the concentrations of NPOC in gas and particle phases (ng m$^{-3}$), respectively; PM is the
measured mass concentration of particulate matter, *i.e.*, PM$_{2.5}$ in this study (μg m$^{-3}$); R is the ideal gas constant (8.314 m$^3$
Pa$^{-1}$ K$^{-1}$ mol$^{-1}$); T is the ambient temperature (K); MW$_{OM}$ is the mean molecular weight (g mol$^{-1}$), and is 200 g mol$^{-1}$ in
this study (Xie et al., 2013); $\xi_{OM}$ is the activity coefficient of each compound in the absorbing phase and assumed to be
unity in this calculation (Xie et al., 2013); P$^o_L$ and P$^o_{L,0}$ are subcooled vapor pressures at T and 298 K, respectively (Pa);
$\Delta H_0$ is vaporization enthalpy of the liquid at 298.15 K. The measured P$^o_L$ and $\Delta H_0$ were extracted from previous
literatures (And and Hanshaw, 2004;Wang et al., 2016).
The total concentration (S) of each NPOC in gas and particle phases was calculated as Eq. (7):
$$S = F + A = \left( 1 + \frac{10^6 \cdot MW_{OM} \cdot \xi_{OM} \cdot P_L^o}{R \cdot T \cdot PM} \right) \times F \qquad (7)$$
Also, Jungle–Pankow model was further used to investigate gas-particle partitioning. In this model, the ratio (φ) of the
concentration of NPOC species in particle phase to the total NPOC concentration was calculated:
$$\varphi = \frac{C_p}{C_p + C_g} = \frac{c \cdot \theta}{c \cdot \theta + P_L^o} \qquad (8)$$





$\log K_p = \log \dfrac{c \cdot \theta}{PM} - \log P_L^o$    (9)
where $\theta$ represents particle surface area per unit volume of air (cm$^2$ cm$^{-3}$) and c is a constant which depends on
thermodynamics of the adsorption process, molecular weight, and surface properties (Pa cm$^{-1}$). In this study, c = 17.2 Pa
cm$^{-1}$, and $\theta$ is $1.1\times10^{-5}$, $1.5\times10^{-6}$ and $4.2\times10^{-7}$ (cm$^2$ cm$^{-3}$) for urban area, rural area and background, respectively.
**3. Results and discussion**
**3.1 Abundance of PM2.5 and NPOCs**
The statistical summary and the abundance of measured PM$_{2.5}$ and NPOC species are shown in Fig.2 and Table S2. The
average PM$_{2.5}$ concentration in all sampling sites was 79.3±37.7 µg m$^{-3}$. The average OC and EC concentrations were
13.8±6.9 and 6.3±2.2 µg m$^{-3}$, respectively. Organic matter (OM) was the most abundant component in PM$_{2.5}$, accounting
for 18.8–27.8% of the total mass, which was estimated to be 1.4 times of OC concentration (Feng et al., 2006;Huang et
al., 2014). Following OM, NO$_3^-$, SO$_4^{2-}$ and NH$_4^+$ were also abundant, accounting for 19.9–22.6%, 16.4–18.1% and
10.4–13.7% of PM$_{2.5}$, respectively.
Fifty-seven NPOCs were identified in this study (Table 2), including 30 n-alkanes, 2 iso-alkanes, 15 PAHs, 5 hopanes
and 5 steranes. Their total concentrations ranged from 31.7 to 388.7 ng m$^{-3}$ (an average of 155.9±55.4 ng m$^{-3}$), accounting
for 0.4–2.4% of OM. This was consistent with the measurement results of NPOCs in Pearl River Delta (PRD) (Wang et
al., 2016) and South China Sea (Zhao et al., 2016) in China, with percentages of 0.1–4.2% and 0.8–1.7%, respectively.
n-Alkanes were the most abundant NPOCs, with concentration of 105.3±55.1 ng m$^{-3}$, accounting for 67.2% of NPOC
concentration. PAHs was the second most abundant species, averagely accounting for 29.2% of NPOC concentration.
Hopanes and steranes were minor constituents, with average concentrations of 3.6±3.0 and 1.8±1.3 ng m$^{-3}$, respectively.
The level of n-alkanes in Jiujiang was comparable to those in Shanghai (Feng et al., 2006) and Hong Kong (Li et al.,
2013), but a bit higher than those in Guangzhou (Xu et al., 2013), British Columbia (Ding et al., 2009) and South China
Sea (Zhao et al., 2016). However, concentrations of PAHs in Beijing and Delhi were 1.7 and 8.2 times higher than that in
Jiujiang, respectively.
**Table 2.** Comparison of NPOC concentrations between Jiujiang City and other areas (ng m$^{-3}$)

|  | n-Alkane [a] | PAHs [b] | Hopane [b] | Sterane [b] | Time | Reference |
|---|---|---|---|---|---|---|
| Jiujiang, China | 105.3±55.1 (16.3–305.0); C$_{11}$–C$_{40}$ | 45.3±17.6 (12.3–96.5); 15 | 3.6±3.0 (0.6–16.4); 5 | 1.8±1.3 (0.3–7.0); 5 | 2016.9–12 | This study |
| Shanghai, China | 32.9–341.9; C$_{17}$–C$_{36}$ | 7.8–151.1; 15 | / | / | 2002–2003 | Feng et al., 2006 |
| Beijing, China | 163.0±193.5; C$_{20}$–C$_{35}$ | 78.7±115.4; 16 | 6.9±6.6; 4 | / | 2004 | Feng et al., 2006 |



| Guangzhou, China | 48.1±20.8; C$_{19}$–C$_{40}$ | 19.0±17.5; 18 | / | / | 2010.11 | Xu et al., 2013 |
|---|---|---|---|---|---|---|
| South China Sea | 15.7–124.2; C$_{15}$–C$_{38}$ | 3.4–127.9; 22 | 0.4–19.6; 8 | 0.4–3.5; 6 | 2013.9–10 | Zhao et al., 2016 |
| Hong Kong, China | 97.9; C$_{29}$–C$_{33}$ | 13.5; 15 | 21.5 | | | Li et al., 2013 |
| PRD$^c$, China | 1.6–436(64); C$_{22}$–C$_{38}$ | 0.1–74(14.2); 17 | 0.3–21(2.3); 10 | 0.01–4.1(3.4);5 | 2011–2012 | Wang et al., 2016 |
| British Columbia Canada | 4.89–74.38(15.58); C$_{11}$–C$_{40}$ | 1.01–41.7 (10.82); 16 | / | / | 2005.12–2007.2 | Ding et al., 2009 |
| Delhi, India | 425±343; C$_{12}$–C$_{35}$ | 373±197; 17 | / | / | 2006–2009 | Yadav et al., 2013 |

ᵃ: mean concentration (concentration range); species;
ᵇ: mean concentration (concentration range); number of species;
ᶜ: including four cities in PRD: Guangzhou, Dongguang, Nanhai and Nansha.

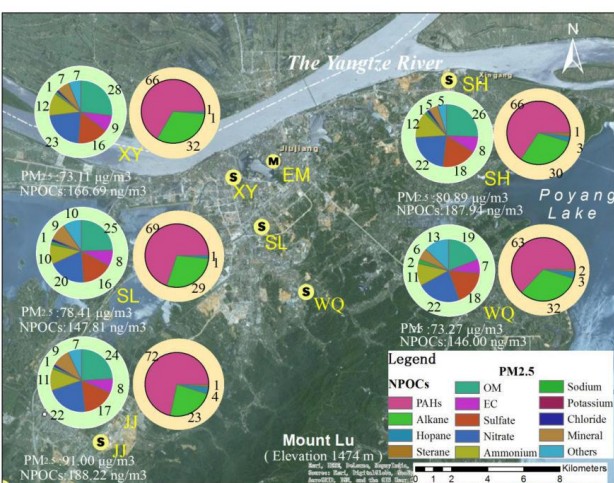

**Fig. 2.** Spatial distributions of NPOCs and PM$_{2.5}$ in Jiujiang city
**3.1.1 PAHs**
PAHs are ubiquitous pollutants of the environment. They are originated from natural and anthropogenic sources such as
biomass burning, vehicle exhausts, residential heating, waste incineration and industrial emissions. It is well established
that the atmospheric concentration and lifetime of PAHs are highly dependent on their gas-particle partitioning behavior,
transformation and degradation. The PM$_{2.5}$-associated PAHs were shown in Fig. 3a. Their individual concentrations
varied between 0.4 and 5.7 ng m$^{-3}$. BbF (5.7 ng m$^{-3}$) was the most abundant PAH species, followed by BaA (5.6 ng m$^{-3}$)
and BaP (4.2 ng m$^{-3}$), together accounting for 34.2% of all PAHs, this was consistent with some previous studies in
Guangzhou (Xu et al., 2013). Because of the strong carcinogenic effect of BaP, special attention should be given to it, the
level of which was also higher than that indicated by the air quality guideline of WHO (1 ng m$^{-3}$).
Because of the vapor pressure dependent partitioning, 2- and 7-ring ring PAHs distributed majored in gas and particle
phases, respectively. However, PAHs with 3–6 rings appeared in both gas and particle phases through gas-particle



partitioning. The total percent contribution of 4- and 5-ring PAHs was 67.9% in this study. FLU–PYR–CHR and
BaA–BaP congeners of the 4-ring PAHs often indicate diesel vehicle and biomass combustion (Yadav et al., 2013),
respectively, and 5-ring BkF is considered as marker of vehicle tracer. This suggests biomass burning and fossil fuel
combustion have mixed effects on local pollution.
MDRs of atmospheric PAHs with similar molecular weight have been widely used as a useful tool for aerosol source
identification. In this study, the ANT/(PHE+ANT) ratios varied in 0.28–0.68 (mean of 0.48), confirming a strong
influence from pyrogenic emissions (Xu et al., 2013;Yadav et al., 2013). Most samples had FLU/(FLU+PYR) ratios of
0.36–0.58, implying combined effects of pyrogenic emission and combustion of fuel, grass, wood and coal (Kuang et al.,
2011;Chen et al., 2016a). The average IcdP/(IcdP+BghiP) ratio was 0.57, and in most cases the ratio > 0.50, suggesting
significant impacts from the combustion of grass, wood and coal. Despite MDRs have been consistently used for source
identification, it remains a rather rough method and contradictions may occur. Therefore, more samples should be
collected to achieve better results.





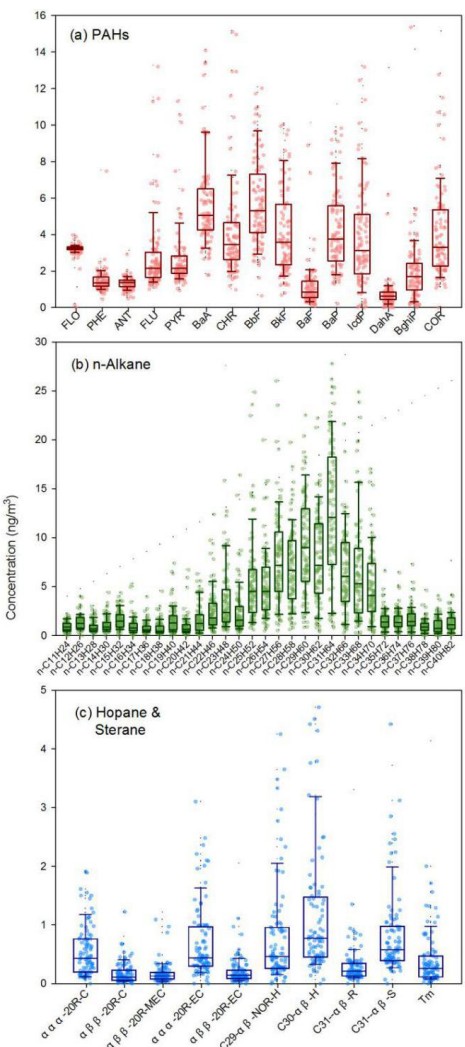


**Fig.3.** Concentration profiles of NPOCs
**3.1.2 n-Alkanes**
Unique signatures of n-alkanes have been shown for different sources, including vehicle exhausts, tire-wear particles,
road dust, cooking oil, cigarette smoke and particulate abrasion products from leaf epicuticular waxes (Rogge et al.,
1994;Ma et al., 2011;Yadav et al., 2013;Zhang et al., 2015). A bimodal distribution of chain lengths of n-alkanes with
peaks at $C_{20}$–$C_{22}$ and $C_{24}$–$C_{27}$ implies vehicle exhaust sources (Zhang et al., 2015). However, a unimodal distribution of
chain lengths of >$C_{30}$ n–alkanes (peak at $C_{37}$) represents tire-wear particle sources. The distribution pattern characterized
by large proportion of $C_{27}$–$C_{33}$ odd n–alkanes suggests vegetation sources.





Our analysis showed that the middle-chain-length n-alkanes ($C_{25}$–$C_{34}$) were the most abundant (Fig. 3b), accounting for
72.3% of the total measured n-alkanes. Feng et al. (2006) and Xu et al. (2013) reported similar findings that $C_{27}$–$C_{29}$
n-alkanes dominated the distribution of n-alkanes in three metropolitan cities of China (Beijing, Shanghai, Guangzhou).
In addition, a predominance of odd carbon numbered congeners ($C_{25}$, $C_{27}$, $C_{29}$, $C_{31}$ and $C_{33}$) was found, with $C_{max}$ = 31 in
most cases and $C_{max}$ = 29 in a few cases (Fig. 3b). The $C_{max}$ value in this study suggests that road dust, cigarette smoke
and tire-wear abrasion products have all contributed to n-alkanes in Jiujiang city. However, no obvious odd/even carbon
preference was observed for either $C_{11}$–$C_{14}$ nor $C_{35}$–$C_{40}$ n–alkanes.
Plant wax n-alkanes exhibit strong odd/even carbon number predominance, while n-alkanes from fossil fuel combustion
do not (Feng et al., 2006). Thus, biogenic n-alkanes should have CPI values greater than unity, whereas anthropogenic
n-alkanes should have CPI values close to unity. Furthermore, CPI < 2 is a typical characteristic of urban environment,
suggesting major contribution from petrogenic sources, e.g., vehicle exhausts and industrial emissions. The CPI values
were 1.00–1.79 (average of 1.29) in this study, implying strong contributions from petrochemical sources, diesel residues
and gasoline emissions. Ding et al. (2009) reported a mean CPI value of 1.5 in central British Columbia of Canada, and
Xu et al. (2013) reported CPI values of 1.2–1.7 (mean of 1.4) in Guangzhou of China. The mean contribution of plant
wax n-alkanes to the total n-alkanes (WNA%) was 17.00±4.41%, ranging from 7.94% to 31.31%. PNA% provides a
direct insight into n-alkanes from petrogenic sources. In this study, PNA% was 83±4.41%, implying that 83% of
n-alkanes were originated from anthropogenic sources. The ACL value varied from 27.45 to 30.95. This small
fluctuations may suggest that emissions were similar at different sites, which was consistent with the result of Delhi in
India (Yadav et al., 2013).
**3.1.3 Hopanes and steranes**
Hopanes and steranes are usually found in crude oil and engine oil, subsequently in vehicle exhausts from unburned
lubricating oil residues. They are regarded as markers of fossil fuel combustion. The concentration profile of hopanes and
alkanes was shown in Fig. 3c. Their total concentrations ranged from 1.1 to 20.5 ng m$^{-3}$, and the concentration of hopanes
was approximately two times of that of steranes. As expected, the total concentrations of hopanes/steranes were 6.7/2.5,
3.3/1.2 and 1.9/1.1 ng m$^{-3}$ in petrochemical industry (SH), traffic area (SL) and suburban area (WQ), respectively.
The predominant hopane analogs were $C_{30}$–$\alpha\beta$–H, $C_{31}$–$\alpha\beta$–S and $C_{29}$–$\alpha\beta$– NOR–H, with concentrations of 1.2±1.3,
0.8±0.7 and 0.8±0.9 ng m$^{-3}$, respectively. The homohopane index ($C_{31}$–S/(S+R)) was 0.75, much greater than those of
diesel (0.49), gasoline vehicle emissions (0.50–0.62) and petroleum (0.6), but a bit smaller than industrial bitumite coal
(0.87) (Fraser et al., 2002). This implies vehicle exhausts, petrochemical emissions and coal combustion have all



contributed to particle concentrations. The concentration profiles of steranes were similar at different sites, with
ααα–20R–EC being the most abundant, followed by ααα–20R–EC. This pattern was a bit different from that in Delhi
which was dominated by $C_{29}$ sterane (Yadav et al., 2013).

**3.2 Size-specific distributions**

Particulate matters within 13 size fractions were collected. The size-specific distribution of NPOCs was then obtained
(Fig. 4). The total concentration of PAHs (Fig. 4a) in each size fraction ranged from 0.4 ng m$^{-3}$ in the 0.01–0.018 μm
fraction to 5.1 ng m$^{-3}$ in the 0.56–1.00 μm fraction. A bimodal distribution of the concentrations of PAHs was observed,
with peaks in 0.56–1.00 and 9.90–18.0 μm fractions, respectively. PAHs in the 0.56–1.00 μm fraction were the most
abundant. This phenomenon could be reasonably explained by that heavy molecular weight PAHs tend to be enriched in
smaller particles (< 1.4 μm) (Kleeman et al., 2008), whereas light molecular weight PAHs are speculated to adsorbed
onto coarse particles by volatilization and condensation. As discussed above, the heavy molecular weight PAHs
accounted for 50.6% of the total PAHs in this study. Similarly, Hien et al. (2007) found that PAHs accumulated
predominantly in small size fractions (especially < 0.4 μm) in urban aerosols. More recently, Mu et al. (2017) indicated
PAHs were strongly correlated with accumulation mode particles (0.05–2.0 μm), and PAHs in this fraction accounted for
~85% of the total measured PAHs. In fact, the relationship between the concentration of PAHs and the size particles is
highly variable. This suggests not only source type but also photodegradation and gas-particle partitioning have great
influences on the size-specific distribution of PAHs.
The concentrations of n-alkanes (Fig. 4b) in Aitken nuclei, Accumulation and Coarse mode particles were 1.7–3.1,
7.0–28.7 and 4.7–6.3 ng m$^{-3}$, respectively. The concentrations of n-alkanes in individual fractions accounted for
1.5–24.5% of the total measured n-alkanes in all fractions. n-Alkanes in the 0.56–1.00 μm fraction were the most
abundant, whereas n-alkanes in three nano-size fractions accounted for the smallest percentages (1.5–2.5%). Notably, the
concentration of n-alkanes increased with increase in fraction size. After fraction size reached 1.00 μm, however, the
concentration of n-alkanes decreased in coarse mode particles. This implies n-alkanes have a tendency to be adsorbed on
fine particles. In general, condensation is more likely happen to fine particles because of their large quantity and larger
specific surface area (Wang et al., 2009).
The size-specific distribution of hopanes and steranes was illustrated in Fig. 4c. Hopanes and steranes were the most
abundant in the following five fractions: 0.56–1.00 μm (2.9 ng m$^{-3}$), 0.32–0.56 μm (2.5 ng m$^{-3}$), 0.18–0.32 μm (1.8 ng
m$^{-3}$), 9.9–18 μm (1.2 ng m$^{-3}$) and 0.10–0.18 μm (1.1 ng m$^{-3}$). Approximately 55% of ∑(hopanes+steranes) were
associated with the 0.10–1.00 μm fraction. This result was consistent with that of Kleeman et al. (2008) who found the



hopanes and steranes were abundant in ultrafine size fraction during a severe winter pollution episode in Sacramento,
USA.

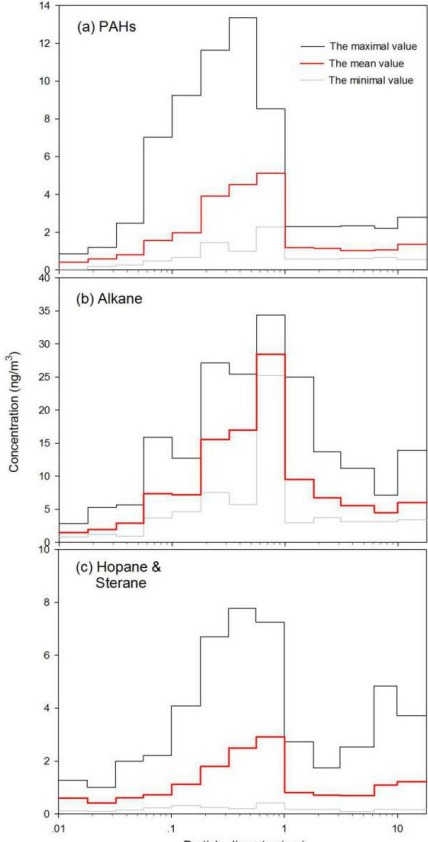


329              **Fig. 4.** Mean-normalized size-specific distribution of NPOCs in the collected PM$_{2.5}$ samples

**3.3 Degradation of organics**
Photochemical oxidation has great influences on the mass concentration and size-specific distribution of NPOCs, and on
their removal and atmospheric fate (May et al., 2012). Ratio–ratio plot was used to visually compare the distributions of
critical marker species and investigate their degradation(Robinson et al. 2006;May et al., 2012). Photochemical decay
could cause the ambient data to be distributed along a line emanating from the source profile, with increasing
photochemical age. The more oxidation, the further ambient ratio is from the source profile. In addition to photochemical
decay, the ratio–ratio plot can also be affected by mixed emissions.
In this study, IcdP/BghiP and C29–αβ–NOR/C30–αβ–H were adopted to explore the degradation of NPOCs (Robinson et
al., 2006;Wang et al., 2016). EC shares common origins with PAHs and hopanes but they are subject to photodegradation.



Two pairs of PAHs and hopanes were normalized by EC. The ratio–ratio plot of IcdP and BghiP normalized by EC from
different sites was depicted in Fig. 5a, together with vehicle (diesel and gasoline) exhausts, tunnel and coal burning
source profiles. Most of the data points were distributed along a line, which overlapped tunnel, vehicle exhaust and
industrial coal, implying ambient hopanes in this city were from vehicle emissions and coal combustion. Wang et al.
(2016) reported that vehicle emissions contributed to atmospheric hopanes in four cities in PRD. There were several
deviation points at left down corner, and the values were smaller than the values of tunnel and vehicle source profile,
indicating mixed influence from traffic origins and degradation. Yu et al. (2011) reported a more apparent liner
distributions of data sets measured in Hong Kong and PRD, which can be attributed to their single vehicle source type. In
Fig. 5b, most of the data were linearly distributed, implying both biomass burning and vehicle emission contribute to
hopanes. Robinson et al. (2006) found hopanes were severely depleted in Pittsburgh, USA, and they attributed this
phenomenon to regional air mass transport affecting the oxidation of condensed-phase organic compounds.

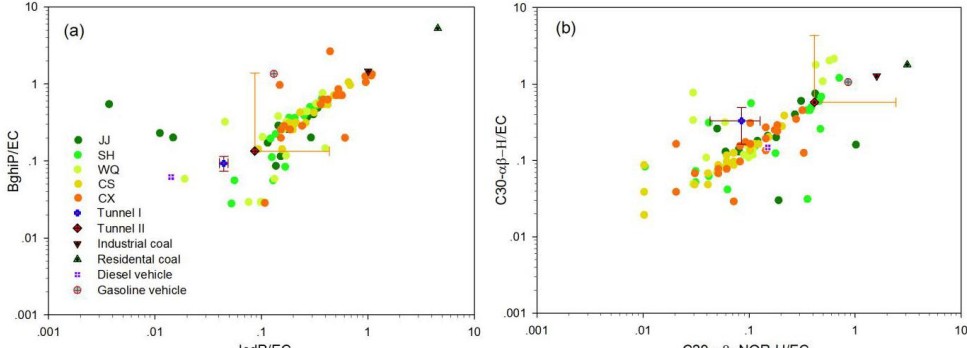


**Fig. 5.** Ratio–ratio plots of two pairs of characterized species (IcdP/BghiP and C29–αβ–NOR–H/C30–αβ–H)
normalized by EC and published source profiles. **(**Tunnel I: Yu et al., 2011; Tunnel II: He et al., 2009; Residental
coal: Zhang et al., 2008; Industrial coal: Zhang et al., 2008; Diesel vehicle: Fraser et al., 2002; Gasoline vehicle:
Fraser et al., 2002**)**
**3.4 Impact of gas-particle partitioning on fine particle source apportionment**
**3.4.1 Gas-particle partitioning**
An important aspect of atmospheric NPOCs is their gas-particle partitioning behavior, which has effects on their fate and
size-specific occurrence. Once NPOCs are emitted into the atmosphere, they subsequently partition between gas and
particle phases and a partitioning equilibrium can be reached according to their vapor pressure and temperature
dependencies.




The particle-phase fraction (φ) of NPOCs was calculated according to the classical gas-particle partitioning model (Fig. 6
and Fig. S1). The gas-phase fractions of LMW PAHs (e.g., FLO, PHE, ANT, FLU and PYR) were rather substantial, and
their particle-phase fractions (φ) ranged from 2.4% to 51.3%. Similarly, the φ values of short chain $C_{22}$–$C_{24}$ n-alkanes
varied between 21.2% and 62.5%, exhibiting an increasing trend with increase in their molecular weight. However, for
the heavier molecular weight species, like PAHs with 5–7 rings, long chain n-alkanes (> $C_{26}$), hopanes and steranes, the φ
values remained greater than 90.0% for all temperature ranges. The calculated φ values of PAHs and n-alkanes were
comparable to those estimated in urban Denver, Chicago and Los Angeles in USA (Xie et al., 2013), but a bit greater than
those in PRD of China (Wang et al., 2016). The lower fractions of NPOCs in gas phase in this study compared with that
in PRD was probably because PRD area is located in a border region between subtropics and tropics. This region has
higher temperature than Eastern China area, especially in cold winter seasons, and the higher temperature can facilitate
the shift of species to gas phase.

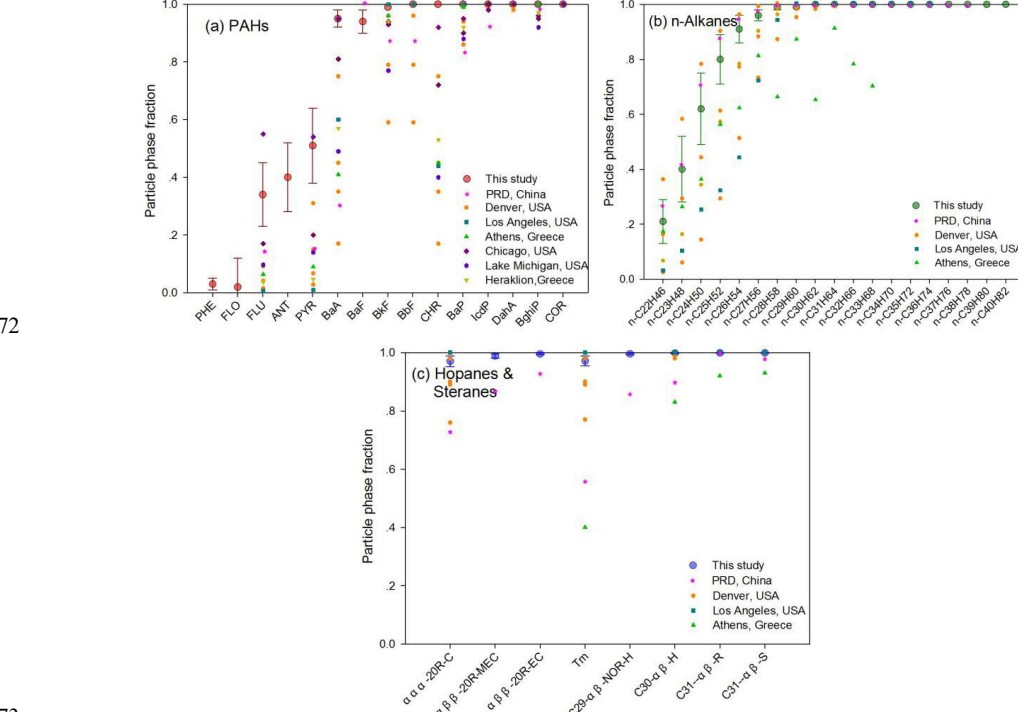

**Fig. 6.** Average particle-phase fractions (φ) of all NPOCs as in comparison with previous results
To further evaluate gas-particle partitioning of NPOCs, φ values were compared with predicted ones by Jungle–Pankow
model (Fig. 7). The φ values of LMW PAHs, short chain n-alkanes, $logP^o_L$>−5 hopanes and steranes, were underpredicted
by Jungle–Pankow model. However, the φ values predicted by Jungle–Pankow model agreed well with the calculated





ones for HMW PAHs and long chain n-alkanes. Underestimation of φ values of PAHs by Jungle–Pankow model
compared with the filed measured ones were also reported by He and Balasubramanian, (2009), and they attributed the
discrepancy to the higher OM fractions in real environment than that adopted by the model.

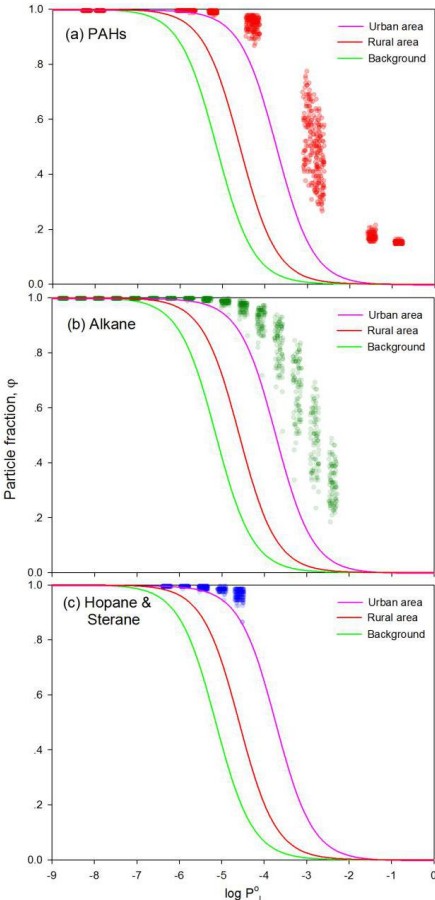


382          **Fig. 7.** Comparison of φ values between the measured and predicted results from Jungle–Pankow model

**3.4.2 PMF source apportionment**
Source apportionment analysis involves techniques that can be used to identify source species and their unique
contributions, which are critical in making policies of controlling pollution. It is typically assumed the molecular markers
are stable in the ambient environment, *i.e.*, being nonreactive and nonvolatile (May et al., 2012). However, many organic
markers can be oxidized over atmospherically relevant time scales, and partition between gas and particle phases. As
discussed above, light molecular weight NPOCs tend to shift to gas phase. Therefore, if the data of NPOCs in single
particle phase are directly used as input for receptor model, this may confound the aerosol factors. To explore the impact



of gas-particle partitioning on PM$_{2.5}$ source apportionment, the NPOCs in single particle phase and the total NPOCs in
gas-particle phases were used as input data for receptor model PMF, respectively. Both input data were incorporated with
elemental species, inorganic ions and OC/EC. Results based on single particle phase and the total phases were denoted by
PMF$_P$ and PMF$_T$, respectively. Five to eleven factors were extracted in this study to obtain reasonable results. Finally, it
turned out that the results of eight factors gave the most reasonable source profiles (Fig. 8 and Fig. S2).
Factor 1 (Fig. 8a) was characterized by significant presence of Al, Ca, Mg, Ti and Fe, which are regarded as good
indicator of dust (including construction dust, geological dust and road dust) (Wang et al., 2015). These elements are the
major elements of dust sand, usually accumulated in the coarse mode particles. Geological dust typically contains high
concentrations of crustal elements, including Fe and Mn. Hence, this factor was regard as "dust", with percent
contributions of 8.90% and 11.0% under PMF$_P$ and PMF$_T$, respectively.
Factor 2 (Fig. 8b) was characterized by the significant presence of Cu, Mn, Zn, Pb, BkF, BbF, BaF and BaP. Mn, Zn and
As are related to emissions from steel production, brick, ceramic and glass making factories (Li et al., 2016;Sulong et al.,
2017). Cu mainly originates from non-ferrous metal production and smelting factories. BkF, BbF and BaP are typical
markers of emissions from coke industry. Several large-scale industrial parks are located in Jiujiang city, e.g., Shacheng
Industry and Jiujiang Comprehensive Industry in the northern and southern areas, respectively. Therefore, factor 2 was
associated with industrial emission.
Factor 3 (Fig. 8c) was characterized by large fractions of HMW PAHs (IcdP, BghiP, DahA and COR), as well as
relatively high fractions of hopanes and steranes. BghiP and COR are excellent tracers of vehicle exhausts. Hopanes and
steranes are related to exhausts from heavy-duty vehicles with diesel engines(Wang et al., 2016). As mentioned above,
there were over 700 thousand motor vehicles in Jiujiang city in 2015, among which about 1/15 were mainly powered by
diesel engines. Therefore, factor 3 was identified as "vehicle related exhausts", with percent contributions of 12.5% and
15.0% under PMF$_P$ and PMF$_T$, respectively.
Factor 4 (Fig. 8d) was characterized by the presence of well-documented indicators of secondary aerosol formation, such
as NO$_3^-$, SO$_4^{2-}$ and NH$_4^+$, with factor fractions of 83.7%, 87.4% and 94.1%, respectively. These secondary products are
formed by precursor gases (SO$_2$ and NOx) via oxidation reactions(Wang et al., 2015). They are mainly emitted from
biomass burning, coal combustion and vehicles. NO$_3^-$, SO$_4^{2-}$ and NH$_4^+$ accounted for 19.9–22.3%, 16.4–18.1% and
10.4–13.7% of PM$_{2.5}$ concentrations, respectively, which were typically derived from gas-particle conversion process as
well as homogeneous and heterogeneous reactions in urban atmosphere. Furthermore, the similar spatial distribution and
contribution in all sites highlight a widespread of these components. Consequently, factor 4 was identified as "secondary
aerosol formation".



Factor 5 (Fig. 8e) was characterized by significant presence of $C_{30}$–$C_{40}$ n–alkanes, as well as relatively significant
presence of FLU and PYR. In previous research (Wang et al., 2016), the long chain n–alkanes ($C_{29}$–$C_{36}$) were considered
to come from local emissions, especially from coal combustion. PYR and FLU are frequently considered as excellent
markers of coal combustion for aerosol source apportionment. Coal is the primary energy source for many industries in
China. About 3.1 million tons of standard coal are consumed per year by the Jiujiang thermal power plant according to
local statistics. Thus, factor 5 was identified as "coal burning", with percent contributions of 18.7% and 16.4% under
$PMF_P$ and $PMF_T$, respectively.
Factor 6 (Fig. 8f) was characterized by high percentage of $Cl^-$ and $K^+$, with some amounts of As, Se, Pb, OC and EC. $Cl^-$
and $K^+$ have been widely used as tracers of wood and biomass burning aerosol (Li et al., 2016). In the past, crop straws
were disposed bu local farmers in the field by burning for convenience. Although this has been extensively banned in
recent years, several large-scale straw burning sites surrounding this city can still be observed by China National Satellite
Meteorological Center (http://hjj.mep.gov.cn/jgjs/). Thus, this factor was considered as "biomass burning", with percent
contributions of 12.7% and 15.7% under $PMF_P$ and $PMF_T$, respectively.
Factor 7 (Fig. 8g) was characterized by high fraction of Ni and V, which are excellent tracers of exhausts from ship and
heavy-duty diesel vehicles. In fact, Jiujiang harbor is among the ten busiest harbors in Yangtze River, whose port cargo
throughput is 59 million tons per year. Hence, factor 7 was identified as "shipping and diesel exhausts".
Factor 8 (Fig. 8h) was characterized by a high load of short chain n–alkanes ($C_{22}H_{46}$, 76.6%; $C_{23}H_{48}$, 84.2%; $C_{24}H_{50}$,
81.1%) and LMW PAHs (about 60% for FLU, PYR, BaA and CHR). These species have several characteristics: most of
their particle-phase fractions ($\varphi$) were less than 50%; relatively light molecular weight; strongly temperature-dependent
vaporization. These compounds have been interpreted as "light NPOCs factor" in previous research (Xie et al.,
2013;Wang et al., 2016). The percent contributions of this factor were 3.7% and 5.6% under $PMF_P$ and $PMF_T$,
respectively. Additionally, concentrations of light NPOCs factor showed an increasing trend with increase in temperature,
implying the association of this factor with fossil fuel evaporation and biogenic emissions. Hence, this factor was
regarded as "light NPOCs factor".





**Fig. 8.** Source profiles of eight sources resolved by PMF




**3.4.3 Assessing impacts of gas-particle partitioning on source apportionment**
As stated above, using the data of single particle phase as input data for PMF model could lead to uncertainty in results,
which was related to gas-particle partitioning of NPOCs in the mathematical solution. In the present study, the eight
extracted factors in this study showed similar source profiles between $PMF_P$ and $PMF_T$. This phenomenon could
reasonably occur because of the major NPOCs compounds were enriched in particle phase in the present study,
meanwhile, the concentrations of hopanes and steranes in gas phase were relatively low. In sharp contrast with this study,
Wang et al. (2016) extracted an extra light NPOCs factor in $PMF_P$ in PRD, and they found very volatile NPOCs (like
FLU) were quite variable in $PMF_T$, but almost did not appear in $PMF_P$. This difference could probably be due to the
$PM_{2.5}$ source identification in this study was focused on the period of high-frequency haze episodes (late autumn and
winter), while the research conducted in PRD included hot summer season which enhanced the uncertainty and
variability of predicted light NPOCs.
Gas-particle partitioning has important effect on the occurrence, atmospheric lifetime and transformation of NPOCs. Also,
size-specific aerosol distribution and photochemical oxidization have influence on gas-particle partitioning. For example,
gas phase oxidation reaction is much faster than heterogeneous reactions, since the uptake of heterogeneous oxidant is
diffusion-limited. The empirical and model predicted particle concentrations would be underestimated or overestimated,
compared with filed measured values. It may add the uncertainty for the input data of receptor model. In this study, using
the data of total organic compounds in gas-particle phases with other aerosol species as input data for receptor model
provides an excellent tool for $PM_{2.5}$ source apportionment. However, reasonable caution should still be given to the more
volatile organic species.
**4. Conclusions**
NPOCs are typical molecular markers for source identification, which attract researchers' interest worldwide. To the best
of our knowledge, this is the first research analyzing size-specific aerosols-associated NPOCs, conducted in a typical
middle-scale city in Eastern China. Fifty-seven $PM_{2.5}$-associated NPOCs including PAHs, n–alkanes, iso–alkanes,
hopanes and steranes were identified and quantified using a TD–GC/MS method. The total concentrations of NPOCs
were 31.7–388.7 ng m$^{-3}$, with n-alkanes being the most abundant species (67.2%). The heavy molecular weight PAHs (4-
and 5-ring) contributed 67.9% of the total PAHs, and the middle chain length ($C_{25}$–$C_{34}$) n-alkanes were the most
abundant in n-alkanes. The MDRs showed that 83.0% of NPOCs were originated from anthropogenic sources, including
pyrogenic source, fossil fuel combustion and biomass burning.



PAHs and n-alkanes were majorly enriched in 0.56–1.00 μm fraction, and ∑(hopanes+steranes) were associated with
0.32–1.00 μm fraction. The nano-size and coarse mode particles showed low concentrations of NPOCs. The ratio–ratio
plots of IcdP/BghiP and C29–αβ–NOR/C30–αβ–H normalized by EC implied that NPOCs in local area were affected by
photochemical degradation and emissions from mixed sources. Gas-particle partitioning model showed that the
calculated particle-phase fraction (φ) of the light molecular weight NPOCs ranged between 2.4% to 62.5%, while those
of heavy PAHs, long chain n-alkanes, hopanes and steranes were high (>90.0%). Using data based on single particle
phase and on gas-particle phases with other $PM_{2.5}$ compounds as input data for PMF model, respectively, we successfully
extracted eight factors from both cases. The $PMF_T$ (the case of gas-particle phases) showed better source profiles than
$PMF_P$ (the case of single particle phase), and the light NPOCs factor contributed a bit more in $PMF_T$ (5.6%) than in $PMF_P$
(3.7%). This study indicates that NPOCs are useful for aerosol apportionment, and total NPOCs in two phases enable
better source profiles than NPOCs in single particle phase.
**Acknowledgement**
This study was financially supported by National Natural Science Foundation of China (No. 21577090) and
Jiujiang Committee of Science and Technology (Grant No. JXTCJJ2016130099). We thank Jiujiang Environmental
Protection Agency and Jiujiang Environmental Monitor Station for coordinating the sampling process and for their
valuable contribution to field measurement. We appreciate senior engineer Ma Yingge (Shanghai Academy of
Environmental Science) and Zou Yajuan (Instrumental Analysis Center, Shanghai Jiao Tong University) for their
assistance in experimental analysis. We also gratefully acknowledge Prof. Yu Jian Zhen (Hong Kong University of
Science and Technology) and Prof. Feng Jialiang (Shanghai University) for their assistance in data processing as well as
many valuable suggestions for improving the manuscript.

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
