# Peer review of "Non-polar organic compounds in aerosols in a typical city of"

_Atmospheric Chemistry and Physics, 2017_

## Referee Comment (RC1) · Anonymous Referee #1 · 5 Feb 2018

This study analyzed the molecular composition of non-polar organic compounds (NPOCs) in PM2.5 and their size distributions at Jiujiang city. The target NPOCs include n-alkanes, PAHs, and hopanes. Diagnostic ratios and PMF model were applied to the compositional data to evaluate the sources and atmospheric processing of PM2.5. In general, this work is well organized and written. However, I still think this work lacks novelty, and would not recommend this manuscript to be accepted for publication at Atmos. Chem. Phys., although a lot of chemical and data analysis work have been done.

General comments:

1. PMF model was utilized to apportion PM2.5 components to factors/sources. However, the author did not provide any information about the method in the manuscript or supporting information. Which version of PMF model (PMF2 or EPA PMF 5.0) was used for source apportionment? How did the author determine the factor number? How did the author deal with the missing values or measurements blow detection limit? Measurement uncertainty was required for PMF input, where were these data from or how were they calculated? Are there any uncertainty analysis related to the PMF modeling? Are the PMF results valid?

2. Page 23, line 449-457. This manuscript stated that the PMFP and PMFT profiles are similar, and should be attributed to the fact that the major NPOCs are enriched in particle phase. This might not apply for factor 6, 7 and 8 (Figure 8f, g and h). The impact of G-P partitioning process will mainly affect the factors highly loaded with low molecular weight species. So the author might need to discuss the impacts of G-P partitioning on these factors.

3. From the title, it seems that the manuscript focused on the size distribution, G-P portioning of NPOCs, and the application of NPOCs on source apportionment of PM2.5. While this study did not measure the gas-phase NPOCs, the gas-particle portioning is only simulated basing on Pankow's theory, and could not be validated. As such, it might not be appropriate to put G-P partitioning in the title, or we can say "G-P partitioning simulation", or "the impacts of G-P portioning on source apportionment". Size distribution was measured for NPOCs, which should be related to other parts of this manuscript. For example, does the size distribution help to explain the PMF results?

4. Diagnostic rations of n-alkanes, PAHs and hopanes were intensively used to evaluate the sources of NPOCs in previous work. The criteria of diagnostic ratios are qualitative and confusing.

Besides the above comments, the lack of enough novelty is the main issue for this work.

[Figure]

The size distribution and diagnostic ratios of NPOCs in typical Chinese cities were intensively investigated (Bi et al., 2005; Zhou et al., 2005; Wang et al., 2009a, b, 2011; Duan et al., 2012). The impacts of G-P portioning of semi-volatile organic compounds (SVOCs) on PMF source apportionment have been observed and validated by Xie et al. (2013, 2014), and the method of using gas + particle phase SVOCs have been intensively applied in PMF source apportionment studies (Gao et al., 2015; Wang et al., 2016; Zhai et al., 2016). Moreover, this work did not provide any new phenomenon or viewpoints that add our knowledge on size distribution or G-P partitioning of NPOCs, or sources apportionment using NPOCs data.

References

Bi, X., Sheng, G., Peng, P.a., Chen, Y., Fu, J., 2005. Size distribution of n-alkanes and polycyclic aromatic hydrocarbons (PAHs) in urban and rural atmospheres of Guangzhou, China. Atmospheric Environment 39, 477-487.

Duan, J., Tan, J., Wang, S., Chai, F., He, K., Hao, J., 2012. Roadside, Urban, and Rural comparison of size distribution characteristics of PAHs and carbonaceous components of Beijing, China. Journal of Atmospheric Chemistry 69, 337-349.

Wang, G., Kawamura, K., Xie, M., Hu, S., Gao, S., Cao, J., An, Z., Wang, Z., 2009b. Size-distributions of n-alkanes, PAHs and hopanes and their sources in the urban, mountain and marine atmospheres over East Asia. Atmos. Chem. Phys. J1 - ACP 9, 8869-8882.

Wang, G., Kawamura, K., Xie, M., Hu, S., Cao, J., An, Z., Waston, J.G., Chow, J.C., 2009a. Organic Molecular Compositions and Size Distributions of Chinese Summer and Autumn Aerosols from Nanjing: Characteristic Haze Event Caused by Wheat Straw Burning. Environmental Science & Technology 43, 6493-6499.

Wang, G., Kawamura, K., Xie, M., Hu, S., Li, J., Zhou, B., Cao, J., An, Z., 2011. Selected water-soluble organic compounds found in size-resolved aerosols collected

from urban, mountain and marine atmospheres over East Asia. Tellus B 63, 371-381.

Wang, Q., Feng, Y., Huang, X., Griffith, S.M., Zhang, T., Zhang, Q., Wu, D., Yu, J.Z., 2016. Nonpolar organic compounds as PM2. 5 source tracers: Investigation of their sources and degradation in the Pearl River Delta, China. Journal of Geophysical Research: Atmospheres 121.

Gao, B., Wang, X.-M., Zhao, X.-Y., Ding, X., Fu, X.-X., Zhang, Y.-L., He, Q.-F., Zhang, Z., Liu, T.-Y., Huang, Z.-Z., Chen, L.-G., Peng, Y., Guo, H., 2015. Source apportionment of atmospheric PAHs and their toxicity using PMF: Impact of gas/particle partitioning. Atmospheric Environment 103, 114-120.

Xie, M., Barsanti, K.C., Hannigan, M.P., Dutton, S.J., Vedal, S., 2013. Positive matrix factorization of PM2.5 - eliminating the effects of gas/particle partitioning of semivolatile organic compounds. Atmospheric Chemistry and Physics 13, 7381-7393.

Xie, M., Hannigan, M.P., Barsanti, K.C., 2014. Impact of Gas/Particle Partitioning of Semivolatile Organic Compounds on Source Apportionment with Positive Matrix Factorization. Environmental Science & Technology 48, 9053-9060.

Zhai, Y., Li, P., Zhu, Y., Xu, B., Peng, C., Wang, T., Li, C., Zeng, G., 2016. Source Apportionment Coupled with Gas/Particle Partitioning Theory and Risk Assessment of Polycyclic Aromatic Hydrocarbons Associated with Size-Segregated Airborne Particulate Matter. Water, Air, & Soil Pollution 227, 44.

Zhou, J., Wang, T., Huang, Y., Mao, T., Zhong, N., 2005. Size distribution of polycyclic aromatic hydrocarbons in urban and suburban sites of Beijing, China. Chemosphere 61, 792-799.

---

## Author Comment (AC1) · 24 Feb 2018

Response to the Interactive comment on "Non-polar organic compounds in aerosols in a typical city of Eastern China: Size distribution, gas-particle partitioning and tracer for PM2.5 source apportionment" by Deming Han et al.
This study analyzed the molecular composition of non-polar organic compounds (NPOCs) in PM2.5 and their size distributions at Jiujiang city. The target NPOCs include n-alkanes, PAHs, and hopanes. Diagnostic ratios and PMF model were applied to the compositional data to evaluate the sources and atmospheric processing of PM2.5. In general, this work is well organized and written. However, I still think this work lacks novelty, and would not recommend this manuscript to be accepted for publication at Atmos. Chem. Phys., although a lot of chemical and data analysis work have been done.

Response: To the best of our knowledge, this study was the first one which systematically researched the size-specific distributions (0.01-18 $\mu$m), photo-degradation and gas-particle partitioning of NPOCs (PAHs, alkane, hopane and sterane), combined diagnostic ratios of special species and receptor model assessing the effect of partitioning on the source apportionment of PM2.5 aerosol. The new information on the profiles of PM2.5-associated NPOCs, size-specific distributions, effect of gas-particle partitioning to the aerosol identification provided by this study, would help us accurately identify the potential sources of aerosols and then asses the contributions from each source.

General comments: 1. PMF model was utilized to apportion PM2.5 components to factors/sources. However, the author did not provide any information about the method in the manuscript or supporting information. Which version of PMF model (PMF2 or EPA PMF 5.0) was used for source apportionment? How did the author determine the factor number? How did the author deal with the missing values or measurements blow detection limit? Measurement uncertainty was required for PMF input, where were these data from or how were they calculated? Are there any uncertainty analysis related to the PMF modeling? Are the PMF results valid?

Response: Considering the limitation of article length, some detailed description was

not added in the original manuscript. According to the suggestion of anonymous Referee 1, the description of PMF analysis would be added in the Section S1 ("PMF analysis and uncertainty assessment") of the revised version of Supplementary material (line 26-52). The detailed added information was as following:

"Positive matrix factorization (PMF) is considered an advanced algorithm among various receptor models, which has been successfully applied for source identification of environmental pollutants. PMF has the following advantages: each data point is given an uncertainty-weighting; the factors in PMF are not necessarily orthogonal to each other and there is no non-negativity constraint with PMF. In the present study, PMF 5.0 (US EPA) was used to apportion the contributions of different sources to PM2.5 in the atmosphere. The matrix X represents an ambient data set in which i represents the number of samples and j the number of chemical species. The goal of multivariate receptor modeling is to identify a number of sources (p), the species profile (f) of each source and the amount of mass (g) contributed by each source to each individual sample as well as the residuals ($e_{ij}$), as equation (S1) (see Fig.1). The PMF solution minimizes the objective function Q based on these uncertainties (u), as equation (S2).

The input data files of PMF consist of concentrations and uncertainty matrices, and the uncertainty data were calculated as equation (S3) as suggested by PMF User Guide. The missing values were represented by average values, while measurements below MDL (method detection limit) were replaced by two times of the corresponding MDL values. The "weak" variables were down-weighted, while "bad" variables were omitted form the analysis process.

The model was run 20 times with 25 random seeds to determine the stability of goodness-of-fit values. It is necessary to test different numbers of sources to find the optimal number of sources which produces the most reasonable results. If the number of sources is estimated properly, the theoretical Q value should be approximately the number of degrees of freedom or the total number of data points. Five to eleven factors were examined, and eight factors were found to be the most appropriate and most reasonably interpretable. Q (True) is the goodness-of-fit parameter calculated including all points, while Q (Robust) is the goodness-of-fit parameter calculated excluding points not fit by the model, Q (Robust) and Q (True) were 1,752.4 and 1,812.9, respectively. Additionally, approximately 98

2. Page 23, line 449-457. This manuscript stated that the PMFP and PMFT profiles are similar, and should be attributed to the fact that the major NPOCs are enriched in particle phase. This might not apply for factor 6, 7 and 8 (Figure 8f, g and h). The impact of G-P partitioning process will mainly affect the factors highly loaded with low molecular weight species. So the author might need to discuss the impacts of G-P partitioning on these factors.

Response: The lower molecule weight species in gas-particle partitioning were more susceptible to influence of the ambient temperature, hence light NPOCs show large fugacity from aerosol surfaces. In the present research, factor 8 was recognized as "Light NPOCs" for the characterization of high load of light NPOCs compounds. However, due to PM2.5 aerosols in this study was mainly conducted in the cold period of high-frequency haze episodes, the resolved factor profiles between PMFP and PMFT model were similar, even for the light NPOCs factor. The discussion of impact gas-particle partitioning on these factor can be seen detailedly in line 448-456 in the Section of "3.4.3 Assessing impacts of gas-particle partitioning on source apportionment" in this original manuscript.

However, for the factor 6 and 7, namely "Biomass burning" and "Shipping and diesel exhaust", which were characterized by inorganic salts (Cl- with K+) and heavy metals (Ni with V), respectively. The tracers for these factors could not be partitioned between phases, despite several light NPOCs species took relative medium to high factor loads, their impacts caused by gas-particle partitioning should be ignored.

3. From the title, it seems that the manuscript focused on the size distribution, G-P portioning of NPOCs, and the application of NPOCs on source apportionment of PM2.5.

While this study did not measure the gas-phase NPOCs, the gas-particle portioning is only simulated basing on Pankow's theory, and could not be validated. As such, it might not be appropriate to put G-P partitioning in the title, or we can say "G-P partitioning simulation", or "the impacts of G-P portioning on source apportionment". Size distribution was measured for NPOCs, which should be related to other parts of this manuscript. For example, does the size distribution help to explain the PMF results?

Response: We still think the original "Non-polar organic compounds in aerosols in a typical city of Eastern China: Size distribution, gas-particle partitioning and tracer for PM2.5 source apportionment" was a proper title, for two main reasons. 1). The size-distribution and gas-particle partitioning of NPOCs was really two major research contents in this study. The gaseous phase of NPOCs for the corresponding 13-staged aerosols were not measured in this study, because of it is still almost impossible of collecting different size-specific aerosol and the corresponding gaseous NPOCs simultaneously. In fact, we adopt the classical gas-particle partition model to simulate the abundance of gaseous NPOCs and explored the particle fraction ($\varphi$) of NPOCs with typical organic matter parameters in urban, rural and background areas. Just as PAHs, alkanes, hopanes and steranes could be called as "NPOCs" in this study, though not all kinds of NPOCs species were analyzed. 2). The NPOCs were used as tracers for the source identification of PM2.5 through specific specie ratios and receptor model, but not source apportionment of NPOCs themselves. If the title was changed to "gas-particle partitioning simulation" or "the impacts of gas-particle partitioning on source apportionment", either it lost the key section of "source identification of PM2.5" or it is wrong for this study.

The size-specific distributions of NPOCs have important influence on their gas-particle partitioning and photo- degradation. Also, combining the characterized species ratios and model extractions, size-specific distributions of NPOCs have relation to aerosols source identifications. The size-distribution of NPOCs was tightly related to the parts of Sections of "Degradation of organics", "Gas-particle partitioning" and "PMF source

apportionment" in this study.

4. Diagnostic rations of n-alkanes, PAHs and hopanes were intensively used to evaluate the sources of NPOCs in previous work. The criteria of diagnostic ratios are qualitative and confusing.

Response: Despite diagnostic ratio was relatively a bit empirical and rough when used for the source identification in some cases, it could easily provide useful information in most situations. Additionally, the combined use of diagnostic ratios of NPOCs and PMF model would provide mutual authentication.

5. Besides the above comments, the lack of enough novelty is the main issue for this work. The size distribution and diagnostic ratios of NPOCs in typical Chinese cities were intensively investigated (Bi et al., 2005; Zhou et al., 2005; Wang et al., 2009a, b, 2011; Duan et al., 2012). The impacts of G-P portioning of semi-volatile organic compounds (SVOCs) on PMF source apportionment have been observed and validated by Xie et al. (2013, 2014), and the method of using gas + particle phase SVOCs have been intensively applied in PMF source apportionment studies (Gao et al., 2015; Wang et al., 2016; Zhai et al., 2016). Moreover, this work did not provide any new phenomenon or viewpoints that add our knowledge on size distribution or G-P partitioning of NPOCs, or sources apportionment using NPOCs data.

Response: China is suffering severe atmospheric pollutions including haze pollution. Due to the heterogeneous development of industrial, economic, geomorphic and environmental conditions, different cities were facing different environmental pressure and situation. Currently, researches of NPOCs were focused on megacities in Eastern China, while most medium cities were ignored. Undoubtedly, systematically analyzing aerosols bound NPOCs and learning their tracers for the source apportionment of PM2.5 in a typical medium city, has great academic and practical values without doubt.

NPOCs as one important class of particles were rather chemically stable, which have been reported by numerous researchers, several researchers use them as tracers for

PM2.5 source apportionment. Just as reviewer suggested (Table C1), there were numerous researches about molecular compositions, sized distributions of organic compounds and their effects on G/P partitioning have been published. However, these researches have their unique research interests, showed different focuses to our manuscript (Table R1), e.g. Wang et al., (2011b), Wang et al., (2009) and Wang et al., (2015) analyzed the concentrations of NPOCs and researched their characterizations. Wang et al., (2011a) reported concentrations, characterization and the size specific (0.4-9.0 $\mu$m) distributions of n-alkanes, PAHs and hopanes in three different typical sites, they neither evaluated their gas-particle partitioning of these compounds, nor investigated their sources and the corresponding contribution. Xie et al., (2014) evaluated the gas-particle partitioning process of six species of PAHs, twelve species of n-alkanes, hopanes and steranes, explored the partitioning impacts on their source apportionment, and got five NPOCs species profiles (odd alkane, light SVOCs, n-alkane, PAHs and sterane).

This manuscript systematically provides new information on the profiles and characterizations of PM2.5-associated NPOCs, evaluated their size-specific distributions and impacts on the gas-particle partitioning, found the effects of gas-particle partitioning and degradation were not apparent o the source apportionment. As best as our knowledge, this is the first research systemically analyzing the characterization, size-specific distribution, gas-particle phase partitioning of NPOCs, and exploring effects of partitioning between tracers for the aerosol source identifications. Based on this manuscript, it will help us to identify the more accurate sources of aerosols and asses the contributions from each source, provide information for further targeted optimized emission control strategies.

Table C1 Comparison between related studies with this manuscript (see Fig.2)

We thank Referee 1 for his good suggestions in the anonymous refereeing process and his/her careful reading our article.

Reference: 1) Wang, G., Kawamura, K., Xie, M., Hu, S., Gao, S., Cao, J., An, Z., Wang, Z., 2009a. Size-distributions of n-alkanes, PAHs and hopanes and their sources in the urban, mountain and marine atmospheres over East Asia. Atmospheric Chemistry and Physics J1 - ACP 9, 8869-8882.

2) Wang, G., Kawamura, K., Xie, M., Hu, S., Cao, J., An, Z., Waston, J.G., Chow, J.C., 2009b. Organic Molecular Compositions and Size Distributions of Chinese Summer and Autumn Aerosols from Nanjing: Characteristic Haze Event Caused by Wheat Straw Burning. Environmental Science Technology 43, 6493-6499.

3) Wang, G., Kawamura, K., Xie, M., Hu, S., Li, J., Zhou, B., Cao, J., An, Z., 2011. Selected water-soluble organic compounds found in size-resolved aerosols collected from urban, mountain and marine atmospheres over East Asia. Tellus B 63, 371-381.

4) Wang, G., Kawamura, K., 2005. Molecular Characteristics of Urban Organic Aerosols from Nanjing:  A Case Study of A Mega-City in China. Environmental Science Technology 39, 7430-7438.

5) Wang, G., Kawamura, K., Lee, S., Ho, K., Cao, J., 2006. Molecular, Seasonal, and Spatial Distributions of Organic Aerosols from Fourteen Chinese Cities. Environmental Science Technology 40, 4619-4625.

6) Gao, B., Wang, X.-M., Zhao, X.-Y., Ding, X., Fu, X.-X., Zhang, Y.-L., He, Q.-F., Zhang, Z., Liu, T.-Y., Huang, Z.-Z., Chen, L.-G., Peng, Y., Guo, H., 2015. Source apportionment of atmospheric PAHs and their toxicity using PMF: Impact of gas/particle partitioning. Atmospheric Environment 103, 114-120.

7) Xie, M., Hannigan, M.P., Barsanti, K.C., 2014. Impact of Gas/Particle Partitioning of Semivolatile Organic Compounds on Source Apportionment with Positive Matrix Factorization. Environmental Science Technology 48, 9053-9060.

8) Huang R J, Zhang Y, Bozzetti C, 2014. High secondary aerosol contribution to particulate pollution during haze events in China. Nature, 514(7521):218.

Deming Han

Tel: +86 21 5474 3936;

Fax: 86 21 5474 0825;

E-mail: handeem@sjtu.edu.cn; jpcheng@sjtu.edu.cn ;

Add.:800 Dongchuan Road, Minhang District Shanghai, China

Room 508, School of Environmental Science and Engineering, Shanghai Jiao Tong University

$$X_{ij} = \sum_{k=1}^{p} g_{ik} f_{kj} + e_{ij} \tag{S1}$$

$$Q = \sum_{i=1}^{n} \sum_{j=1}^{m} \left[ \frac{X_{ij} - \sum_{k=1}^{p} g_{ik} f_{kj}}{u_{ij}} \right]^{2} \tag{S2}$$

$$\begin{cases} Unc_i = \frac{5}{6} \times MDL_i & C_i \leq MDL_i \\ Unc_i = \sqrt{(C_i \times Error\ Fraction)^2 + \left(\frac{1}{2} \times MDL_i\right)^2} & C_i > MDL_i \end{cases} \tag{S3}$$

**Fig. 1.**

**Table C1** Comparison between related studies with this manuscript

| ID | Studies | Sampling site | Sampling duration | Analysis method | PAHs | n-Alkanes | Hopanes | Steranes | Size-distribution | Gas-particle partitioning | Source identification | Remark |
|---|---|---|---|---|---|---|---|---|---|---|---|---|
| ① | Wang et al., 2011a; ACP | Baoji city, China | 11-14, Jan. 12-20, Feb. 12-24 Apr. 2008 | Ultrasonication Extraction+GC/MS | 16 species | C18-C36 | 5 species | / | TSP; 0.4-9.0 µm, nine stage; | / | / | |
| | | Mount Tai, China | 22-29 Jun. 2006; 12-24 Jan. 2008 | | | | | | | | | |
| | | Okinawa Island, Japan | 18 Mar. -12 Apr. 2008 | | | | | | | | | |
| ② | Wang et al., 2011b; EST | Nanjing city, China | 1-17 Jan. 2007 12-14 Oct. 2007 | Ultrasonication Extraction+GC/MS | 16 species | C18-C34 | / | / | 0.4-9.0 µm, nine stage; | / | / | |
| ③ | Wang et al., 2009; Tellus, B | The same to study ① | The same to study ① | Ultrasonication Extraction+GC/MS | / | / | / | / | 0.4-9.0 µm, nine stage; | / | / | Mainly about sugar, sugar-alcohol, carbonxylic acid |
| ④ | Wang et al., 2015; EST | Nanjing city, China | Jul. 2004 - Jan. 2005 | Solvent extraction+ GC/MS | / | C18-C36 | / | / | / | / | Tracers for source identification | |
| ⑤ | Wang et al., 2006; EST | 14 Chinese cities | 2 days in winter + 2 days in summer, 2003 | Ultrasonication Extraction+GC/MS | 18 species | C16-C35 | C27-C32 | / | / | / | / | Other sugars and so on |
| ⑥ | Gao et al., 2015; AE | Guizhou city, China | 28 Nov. - 23 Dec. 2009 | Ultrasonication Extraction+GC/MS | 13 species | / | 4 species | / | / | / | Factor identification + correlation analysis | |
| ⑦ | Xie et al., 2014; EST | Denver, USA | Aug. 2012- Jul. 2013 | / | 6 species | 12 species | 5 species | 5 species | / | / | PMF model | Extracted five species profiles |
| | This manuscript; ACP | Jiujiang city, China | Sep. - Dec. 2016 | TD-GC/MS, without solvent extraction | 15 species | 30species, C11-C40 | 5 species | 5 species | 13 stage, 0.01-18 µm | Partitioning between all these NPOCs | Tracers + PMF model | Extracted 8 factors for PM$_{2.5}$ aerosols |

**Fig. 2.**

---

## Referee Comment (RC2) · Deming Han et al. · 28 Mar 2018

The manuscript has the potential to add to the available body of evidence. This work
details the size distribution, gas-particle partitioning and source apportionment of air-
borne PM2.5-associated non-polar organic compounds in one city of eastern China. In
general, I recommend that the manuscript be accepted pending some major revisions
as outlined below.

1. How about the air quality of Jiujiang City? Is there any public data about the air
quality there?

[Figure]

2. Please cite the references at the end of the sentence, not in the middle.

3. Line 97, please give the size of the quartz fiber filter used. And how was the air flow rate of high-volume air sampler? Detailed information should be given.

4. Line 98, did you mean that each sample was performed for five continuous days? Did you consider the filter would be oversaturated for such long sampling time?

5. How many samples did you get totally?

6. Line 107, why did you choose two sizes of quartz fiber filters? Was one for high-volume air sampler, while other for MOUDI? Modify clearly and suitably.

7. Line 129, why didn't you choose SIM mode?

8. Line 143, "PM 2.5 samplers were placed on the rooftop at EM site with distance between any two samplers <3 m, collecting for 12 h, then the added mass was calculated." This sentence is not written in proper English, and it's not clear what point is being made here. I recommend deleting this sentence.

9. When did you add the internal standards?

10. Should give some short information on SRM 2260, SRM 2260A and SRM1494.

11. I can't find you any statement about the recovery. Did you subtract the average blank level from all samples? Please clarify.

12. Please state what the levels were in the processing blanks, and how thy compared to your lowest standard. How do you define the LOQ?

13. Did you include spike blanks? How was accuracy of the method evaluated?

14. Line 215, is it your own data or cited data from reference? How did you determine OM?

15. Figure 4, the image resolution is poor.

16. Line 333-335, References should be cited.

17. Line 406, Line 362, Line 400, the abbreviations of the PAH compounds should be given the full expressions.

18. Authors should give some reasons for introduced elemental species, inorganic ions and OC/EC into PMF model.

---

## Author Response (AR1)

**Dear Editors and Reviewers:**

Thank you for your letter and for the reviewers' comments concerning our manuscript entitled "Non-polar organic compounds in aerosols in a typical city of Eastern China: Size distribution, gas-particle partitioning and tracer for $PM_{2.5}$ source apportionment" (Ref: acp-2017-908). These comments are valuable and very helpful for revising and improving our paper, as well as the important guiding significance to our researches. We have studied comments carefully and have made correction, the correction in the manuscript was marked-up with GREEN colour which we hope meet with approval. The main corrections in the paper and the responds to the reviewer's comments are as flowing:

Responds to the editors' and reviewers' comments:

**Anonymous Referee #3**

**The manuscript has the potential to add to the available body of evidence. This work details the size distribution, gas-particle partitioning and source apportionment of airborne PM2.5-associated non-polar organic compounds in one city of eastern China. In general, I recommend that the manuscript be accepted pending some major revisions as outlined below.**

**1. How about the air quality of Jiujiang City? Is there any public data about the air quality there?**

*Response:* There was a small number of published researches about the atmosphere quality of Jiujiang City. According to Yang's (2016) research, $PM_{2.5}$ was the dominant atmospheric pollutant in Jiujiang City, which ranged from 31 to 196 $\mu g\ m^{-3}$ in 2014-2015, and the highest abundances of aerosols was usually occurred during October to January of the next year.

According to a recent research program "Source Apportionment of $PM_{2.5}$ and VOCs, and Control and Management of Jiujiang City" (data not published), the average concentrations of $PM_{2.5}$ and $O_3$ in Jiujiang City were 45, 51, 50 $\mu g\ m^{-3}$ and 43, 55 and 61 $\mu g\ m^{-3}$ in 2014–2016, respectively. $PM_{2.5}$ and $O_3$ both showed a gradual rising trend, with annual growth rate of 4.3% and 20.9%, respectively. As for chemical constituents, the secondary inorganic aerosols occupied 53.4% of the $PM_{2.5}$ concentration, while OC occupied of 22.1%. Factors extracted by PMF, CMB receptor models showed that coal combustion, vehicle exhaust, industrial production (petrochemical) and dust contributed 31.4%, 14.3%, 13.1% and 11.2%, respectively. The $\Sigma_{103}$VOCs (volatile organic compounds) concentration was 11.5-197.8 ppbv, with average value of 78.20±49.92 ppbv, which is close to in megecities of Beijing and Shanghai in China. Aromatic hydrocarbon was the primary VOCs component, which occupied of 34.6% of the total VOCs. The source identification

of VOCs extracted by PMF showed the largest contributor was the solvent/coating emission (21.1%), followed by incomplete combustion which accounting for 19.8%.

**2. Please cite the references at the end of the sentence, not in the middle.**

*Response:* According to referee's suggestion, nearly all these references cited in the middle were revised as be cited at the end of the corresponding sentence in the revised manuscript. For example, line 30-31 "In recent years, severe atmospheric pollution characterized by haze has been occurring in developing countries (Yadav et al., 2013;Wang et al., 2015), affecting visibility, optical radiation and human health (Shen et al., 2015;Sulong et al., 2017). " was changed to "In recent years, severe atmospheric pollution characterized by haze has been occurring in developing countries, affecting visibility, optical radiation and human health (Yadav et al., 2013;Shen et al., 2015;Wang et al., 2015;Sulong et al., 2017). " in lines 30-31 in the revised manuscript.

However, several references were kept cited in the middle of the sentence due to the special structure of sentence. For example, in line 243-244 "This was consistent with the measurement results of NPOCs in Pearl River Delta (PRD) (Wang et al., 2016) and South China Sea (Zhao et al., 2016) in China, with percentages of 0.1–4.2% and 0.8–1.7%, respectively." was not changed.

**3. Line 97, please give the size of the quartz fiber filter used. And how was the air flow rate of medial-volume air sampler? Detailed information should be given.**

*Response:* According to referee's suggestion, the size of particle filter and sampler flow were added to the corresponding descriptions in line 97 of the original manuscript, changed as "All $PM_{2.5}$ filter samples were collected using medial-volume air samplers (YH-5, Qingdao, China), at a flow rate of 100 L $min^{-1}$. Particles were collected on quartz fiber filters (GE Whatman, 1851-090, England, UK) with a diameter of 90 mm, each for a duration of 23 h." in lines 98-100 in the revised manuscript.

**4. Line 98, did you mean that each sample was performed for five continuous days? Did you consider the filter would be oversaturated for such long sampling time?**

*Response:* According to referee's suggestion, the statements the sampling duration 23 h for each single sample was added, and in general the filter would not be over-saturated for such a sampling time in this study. The revised description was "Particles were collected on quartz fiber filters (GE Whatman, 1851-090, England, UK) with a diameter of 90 mm, each for a duration of 23 h." in lines 99-100 in the revised manuscript.

**5. How many samples did you get totally?**

*Response:* According to referee's suggestion, the total sampling number of PM2.5 was added in the revised manuscript, as "A total of 137 $PM_{2.5}$ valid samples were collected, as 18 samples were invalid or missing due to the bad weather or power problems." in lines 103-104 in the revised manuscript.

**6. Line 107, why did you choose two sizes of quartz fiber filters? Was one for medial- volume air sampler, while other for MOUDI? Modify clearly and suitably.**

*Response:* This MOUDI sampler was designed to collect particles with diameters of 0.056-18.0 and 0.010-0.056 μm using 47 and 90 mm filters, respectively. According to referee's suggestion, line107 "Two kinds of quartz fiber filters (diameters of 47 and 90 mm,respectively) were prebaked at 550 ℃ for 4 h," was changed to "Two kinds of quartz fiber filters of diameters of 47 and 90 mm were used to collect particles with diameter of 0.056-18.0 and 0.010-0.056 μm, respectively. All the filters were prebaked at 550 ℃ for 4 h," in lines 112-113 of the revised manuscript.

**7. Line 129, why didn't you choose SIM mode?**

*Response:* The TD-GC/MS method used for the aerosol associated NPOCs analysis in this study, was referred to studies conducted by Ho and Yu (Ho and Yu, 2004; Ho et al., 2008), and with several parameter be modified. Numerous previous researches which studied NPOCs via using TD-GC/MS selected SCAN mode instead of SIM mode, see following Table R1.

**Table R1**   References related to scan mode for NPOCs by TD-GC/MS approach

| Location | Instrument | SCAN or SIM | Target compounds | Reference |
|---|---|---|---|---|
| British Columbia, Canada | TDU+Agilent GC6890/MS5973 | SCAN: 50-1000 amu | PAHs, n-alkanes, biomarker | Ding et al., 2009 |
| Hong Kong, China; Fresno, CA, USA | TDU+Agilent GC6890/MS5973 | SCAN: 50-650 amu | PAHs, n-alkanes, iso/anteiso-alkanes, hopanes, steranes, branched alkanes, cyclohexanes, alkenes, pathalates | Ho et al., 2008 |
| Hong Kong, China; | TDU+Agilent GC5890/MS5791 | SCAN: 50-650 amu | PAHs, n-alkanes, | Ho and Yu 2004 |
| Six cities in China and Japan | TDU+Agilent GC6890/MS5975 | SCAN: 50-650 amu | PAHs, n-alkanes, hopanes, steranes, pathalates | Ho et al., 2011 |
| Delhi, India | TDU+Shimadzu GC/MS QP2010 Plus | SCAN: 40-900 amu | PAHs, n-alkanes | Yadav et al., 2013 |
| Pearl River Delta, China | TDU+Agilent GC/MS | SCAN: 50-650 amu | PAHs, n-alkanes, hopanes, steranes | Wang et al., 2016 |
| Jiujiang, China | TDU + Shimadzu GC/MS QP2010 Plus | SCAN: 50-500 amu | PAHs, n-alkanes, hopanes, steranes | This study |

Accordingly, the description of scan mode and determining NPOCs species was revised as "The MS was operated in scan mode with mass range was m/z 50–500, and scanned at 0.5 s/scan (Ho et al., 2008;Yadav et al., 2013). The ion was produced from electronic impact ionization (EI) at 70 eV, and then was separated by high performance quadrupole mass filter. Species identification was achieved via comparing the mass spectra and retention times of the chromatographic peak with the corresponding authentic standards." in lines 140-143 of the revised manuscript.

**8. Line 143, "PM$_{2.5}$ samplers were placed on the rooftop at EM site with distance between any two samplers <3 m, collecting for 12 h, then the added mass was calculated." This sentence is not written in proper English, and it's not clear what point is being made here. I recommend deleting this sentence.**

*Response:* According to referee's suggestion, the sentence in line 143 was shortened and revised as "PM$_{2.5}$ samplers were placed on the building rooftop of Jiujiang Environmental Monitor Station, each two within distance of <3 m." in line 160-161 in the revised manuscript.

**9. When did you add the internal standards?**

*Response:* The internal standards were added to the sample after sample be added with glass-wool plugs. According to referee's suggestion, "The internal standards of n–tetracosane d$_{50}$ (n–C$_{24}$D$_{50}$), naphthalene–d$_8$, acpnaehthene–d$_{10}$, phenanthrene–d$_{10}$, and chrysene–d$_{12}$ were spiked into each sample, through a pipette with a long thin tip. This was done to account for the loss of components from sample filters associated with the instrument instability due to changes in laboratory environmental conditions. After the evaporating of solvent from internal standard was conducted via air drying for several seconds, the TD tubes were capped and put into a sampler holder. " in lines 128-133 in the revised manuscript

**10. Should give some short information on SRM 2266, SRM 2260A and SRM1494.**

*Response:* According to referee's suggestion, the description of NPOCs standards were revised as "The NPOCs standards used were National Institute of Standards and Technology (NIST, USA) Standard Reference Materials (SRM), including SRM 2260A, SRM 1494 and SRM 2266 for 35 PAHs, 30 n-alkanes and 10 hopanes/steranes, respectively." in lines 165-167 in the revised manuscript.

**11. I can't find you any statement about the recovery. Did you subtract the average blank level from all samples? Please clarify.**

*Response:* According to referee's suggestion, the statement of recovery was added in this revised manuscript, depicted

as "Recovery experiment were conducted to improve the desorption of targeted compounds from filters and experimental detection. The analytical recovery was calculated via spiking a known amount of the solution to blank filter, and most compounds were recovered with recovery efficiency >90% except for several light molecule weight species. " in lines 177-1780 in the revised manuscript.

The blank experiments conducted in this study were composed of filed blank, transport blank and laboratory blank experiments, and the final reported data were subtracted the average blank results, just as depicted "Field blanks were collected by keeping blank filters in the sampler for the same duration at sampling site. Additionally, both transport and laboratory blank filters were analyzed, and all the data reported in this study were corrected according to the results." in line 161-164 in the revised manuscript.

**12. Please state what the levels were in the processing blanks, and how thy compared to your lowest standard. How do you define the LOQ?**

*Response:* Processing blank is processed through some or all equipment used for collecting and processing environmental samples. We have conducted processing blank experiments in this study, results showed that the contamination in the processing period was negligible, which may be due to the experiment were conducted in the clean laboratory room. The lowest standard values were compared with these blank values, which met related requirement. The limit of quantification (LOQ) was calculated as ten times values of of S/N (signal/noise).

**13. Did you include spike blanks? How was accuracy of the method evaluated?**

*Response:* We have not conducted spike blank experiments in this study, due to their similar functions to processing blanks. Despite spike blanks were not included, filed blanks, laboratory blanks and processing blanks all were conducted and received rather good results, which suggesting our experiment was receivable.

The description of the accuracy of the method was added in the revised manuscript, depicted as "The accuracy of the method was evaluated by reproducibility of the standard and selected samples ascertained by processing in quintuplicate, and results suggest the analytical precision was better than 5%." in line 180-181 in the revised manuscript.

**14. Line 215, is it your own data or cited data from reference? How did you determine OM?**

*Response:* The number used in line 215 "Organic matter (OM) was the most abundant component in $PM_{2.5}$, accounting for 18.8–27.8% of the total mass" was our own data which calculated by the measured OC in Jiujiang city. Considering

referee's suggestion, this statement was changed to "Organic matter (OM) was the most abundant component in PM2.5, accounting for 18.8–27.8% of the total mass in this study" in lines 237-238 in the revised manuscript.

According to previous researches which studying OM concentrations in Shanghai in YRD area, OM was calculated as OM=1.4×OC, just as "which was estimated to be 1.4 times of OC concentration (Feng et al., 2006; Huang et al., 2014)." in lines 238-239 in the revised manuscript.

**15. Figure 4, the image resolution is poor.**

*Response:* According to referee's suggestion, the Figure 4 was replaced by another figure with high resolution.

[Figure]

**Fig. 4.** Mean-normalized size-specific distribution of NPOCs in the collected PM2.5 samples (Left for the original figure, Right for the revised figure)

**16. Line 333-335, References should be cited.**

*Response:* According to referee's suggestion, references were cited in this revised manuscript, revised as "Photochemical decay could cause the ambient data to be distributed along a line emanating from the source profile, with

increasing photochemical age (Robinson et al., 2006;Yu et al. 2011). " in lines 355-357 in the revised manuscript.

**17. Line 406, Line 362, Line 400, the abbreviations of the PAH compounds should be given the full expressions.**

*Response:* According to referee's suggestion, the abbreviations of PAHs were given in full expressions. And the abbreviation, $P^o_L$ and $\Delta H_0$ information for individual NPOCs in Table S1 in the Supplementary Material, was changed to the main manuscript as Table 2.

**Table 2.** Abbreviation, $P^o_L$ and $\Delta H_0$ information for individual NPOCs

| Species | Abb. | $P^o_L$ [a] | $\Delta H_0$ [b] | Species | Abb. | $P^o_L$ | $\Delta H_0$ |
|---|---|---|---|---|---|---|---|
| **PAHs** | | | | **n–Alkanes** | | | |
| Fluorene | FLO | 1.10E–01 | 84.9 | n–Undecane | $C_{11}H_{24}$ | / | / |
| Phenanthrene | PHE | 2.57E–02 | 88.9 | n–Dodecane | $C_{12}H_{26}$ | / | / |
| Anthracene | ANT | 1.21E–03 | 99.7 | n–Tridecane | $C_{13}H_{28}$ | / | / |
| Fluoranthene | FLU | 1.60E–03 | 98.3 | n–Tetradecane | $C_{14}H_{30}$ | / | / |
| Pyrene | PYR | 7.60E–04 | 97.9 | n–Pentadecane | $C_{15}H_{32}$ | / | / |
| Benz[a]anthracene | BaA | 3.45E–05 | 108 | n–Hexadecane | $C_{16}H_{34}$ | / | / |
| Chrysene | CHR | 1.36E–06 | 118.8 | n–Heptadecane | $C_{17}H_{36}$ | / | / |
| Benzo[b]fluoranthene | BbF | 1.00E–06 | 119.2 | n–Octadecane | $C_{18}H_{38}$ | / | / |
| Benzo[j+k]fluoranthene | BkF | 4.66E–06 | 113 | n–Nonadecane | $C_{19}H_{40}$ | / | / |
| Benzo[a]fluoranthene | BaF | 4.66E–05 | 113 | n–Eicosane | $C_{20}H_{42}$ | / | / |
| Benzo[e]pyrene | BeP | 7.89E–07 | 117.9 | n–Heneicosane | $C_{21}H_{44}$ | / | / |
| Indeno[1,2,3-cd]pyrene | IcdP | 1.42E–06 | 124 | n–Docosane | $C_{22}H_{46}$ | 3.24E–03 | 115 |
| Dibenz[a,h]+[a,c]anthracene | DahA | 4.93E–09 | 134.1 | n–Tricosane | $C_{23}H_{48}$ | 1.22E–03 | 120 |
| Benzo[ghi]perylene | BghiP | 1.01E–08 | 129.9 | n–Tetracosane | $C_{24}H_{50}$ | 4.66E–04 | 124 |
| Coronene | COR | 3.56E–10 | 143.2 | n–Pentacosane | $C_{25}H_{52}$ | 1.72E–04 | 129 |
| **iso–Alkane** | | | | n–Hexacosane | $C_{26}H_{54}$ | 6.59E–05 | 133 |
| Pristane | $C_{19}H_{40}$ | / [c] | / | n–Heptacosane | $C_{27}H_{56}$ | 2.53E–05 | 137 |
| Phytane | $C_{20}H_{42}$ | / | / | n–Octacosane | $C_{28}H_{58}$ | 9.42E–06 | 142 |
| **Hopane** | | | | n–Nonacosane | $C_{29}H_{60}$ | 3.55E–06 | 146 |
| αβ–Nnorhopane | $C_{29}$–αβ–NOR–H | 2.74E–06 | 126 | n–Triacontane | $C_{30}H_{62}$ | 1.32E–06 | 151 |
| αβ–Hopane | $C_{30}$–αβ–H | 1.01E–06 | 130 | n–Hentriacontane | $C_{31}H_{64}$ | 4.96E–07 | 155 |
| αβ–22R–Homohopane | $C_{31}$—αβ–R | 3.85E–07 | 134 | n–Dotriacontane | $C_{32}H_{66}$ | 1.93E–07 | 160 |
| ab 22S–Homohopane | $C_{31}$—αβ–S | 3.85E–07 | 134 | n–Tritriacontane | $C_{33}H_{68}$ | 7.09E–08 | 164 |
| 22,29,30–Trisnorhopane | Tm | 1.93E–05 | 117 | n–Tetratriacontane | $C_{34}H_{70}$ | 2.63E–08 | 169 |
| **Sterane** | | | | n–Pentatriacontane | $C_{35}H_{72}$ | 1.00E–08 | 173 |
| ααα–20R Cholestane | ααα–20R–C | 2.03E–05 | 121 | n–Hexatriacontane | $C_{36}H_{74}$ | 3.75E–09 | 177 |
| αββ–20R Cholestane | αββ–20R–C | / | / | n–Heptatriacontane | $C_{37}H_{76}$ | 1.42E–09 | 182 |
| αββ–20R24S–Methylcholestane | αββ–20R–MEC | 7.60E–06 | 125 | n–Octatriacontane | $C_{38}H_{78}$ | 5.37E–10 | 186 |
| aaa 20R24R–Ethylcholestane | ααα–20R–EC | / | / | n–Nonatriacontane | $C_{39}H_{80}$ | 2.03E–10 | 191 |
| αββ–20R24R–Ethylcholestane | αββ–20R–EC | 2.84E–06 | 130 | n–Tetracontane | $C_{40}H_{82}$ | 7.60E–11 | 195 |

[a]: pure compound vapor pressure, unit of Pa at 298 K, cited from And and Hanshaw, 2004, Xie et al., 2013;

[b]: vaporization enthalpy, unit of (KJ mol$^{-1}$) at 298 K, cited from Xie et al., 2013, Wang et al., 2016;

[c]: "/" means lack of related data.

**18. Authors should give some reasons for introduced elemental species, inorganic ions and OC/EC into PMF model.**

*Response:* According to referee's suggestion, the reasons for introduced elemental species, inorganic ions and OC/EC into PMF model were added in the revised manuscript. The added statement was "Additionally, chemical constituent of ambient aerosol is an essential step to identifying major sources and quantifying the corresponding contributions to particulate matter. Individual organic tracers, elemental species, inorganic ions and OC/EC have been demonstrated to be able to provide source apportionment of aerosols. " in lines 414-416 in the revised manuscript.

**Special thanks to Referee #3 for his/her good comments and his careful reading our manuscript!**

**Reference:**

Ding, L. C., Ke, F., Wang, D. K. W., Dann, T., & Austin, C. C. (2009). A new direct thermal desorption-gc/ms method: organic speciation of ambient particulate matter collected in Golden, BC. Atmos. Environ., 43(32), 4894-4902.

Ho, S. S., Yu, J. Z., Chow, J. C., Zielinska, B., Watson, J. G., & Sit, E. H., et al. (2008). Evaluation of an in-injection port thermal desorption-gas chromatography/mass spectrometry method for analysis of non-polar organic compounds in ambient aerosol samples. J. Chromatography A, 1200(2), 217-27.

Ho, S. S., & Yu, J. Z. (2004). In-injection port thermal desorption and subsequent gas chromatography-mass spectrometric analysis of polycyclic aromatic hydrocarbons and n-alkanes in atmospheric aerosol samples. J. Chromatography A, 1059(1–2), 121-129.

Ho, S. S. H., Chow, J. C., Watson, J. G., Ng, L. P. T., Kwok, Y., & Ho, K. F. (2011). Precautions for in-injection port thermal desorption-gas chromatography/mass spectrometry (TD-GC/MS) as applied to aerosol filter samples. Atmos. Environ., 45(7), 1491-1496.

Wang, Q., Feng, Y., Huang, X. H. H., Griffith, S. M., Zhang, T., Zhang, Q., Wu, D., and Yu, J. Z. (2016). Non-polar organic compounds as PM2.5 source tracers: Investigation of their sources and degradation in the Pearl River Delta, China, J. Geophy. Res. Atmos 121, 11862-11877.

Yadav, S., Tandon, A., and Attri, A. K. (2013). Characterization of aerosol associated non-polar organic compounds using TD-GC-MS: a four year study from Delhi, India, J. Hazard. Materials, 252–253, 29-44.

Yang C., Xu J., Li H. Hong Y. (2016). Temporal variation of $PM_{2.5}$ and its relationship with meteorological conditions in Jiujiang City. Jiangxi Science, 34(6):790-794. (In Chinese);

**Anonymous Referee #1**

**This study analyzed the molecular composition of non-polar organic compounds (NPOCs) in PM$_{2.5}$ and their size distributions at Jiujiang city. The target NPOCs include n-alkanes, PAHs, and hopanes. Diagnostic ratios and PMF model were applied to the compositional data to evaluate the sources and atmospheric processing of PM$_{2.5}$. In general, this work is well organized and written. However, I still think this work lacks novelty, and would not recommend this manuscript to be accepted for publication at Atmos. Chem. Phys., although a lot of chemical and data analysis work have been done.**

*Response:* To the best of our knowledge, this study was the first one which systematically researched the size-specific distributions (0.01-18 μm), photo-degradation and gas-particle partitioning of NPOCs (PAHs, alkane, hopane and sterane), combined diagnostic ratios of special species and receptor model assessing the effect of partitioning on the source apportionment of PM$_{2.5}$ aerosol. The new information on the profiles of PM$_{2.5}$-associated NPOCs, size-specific distributions, effect of gas-particle partitioning to the aerosol identification provided by this study, would help us accurately identify the potential sources of aerosols and then asses the contributions from each source.

**General comments:**

**1. PMF model was utilized to apportion PM$_{2.5}$ components to factors/sources. However, the author did not provide any information about the method in the manuscript or supporting information. Which version of PMF model (PMF2 or EPA PMF 5.0) was used for source apportionment? How did the author determine the factor number? How did the author deal with the missing values or measurements blow detection limit? Measurement uncertainty was required for PMF input, where were these data from or how were they calculated? Are there any uncertainty analysis related to the PMF modeling? Are the PMF results valid?**

*Response:* Due to the limitation of article length, some detailed description was not added in the original manuscript. According to reviewer's suggestion, the description of PMF analysis would be added in the Section S1 *PMF analysis and uncertainty assessment* of the revised version of Supplementary material (line 27-53). The detailed information was as following:

"Positive matrix factorization (PMF) is considered an advanced algorithm among various receptor models, which has been successfully applied for source identification of environmental pollutants (Han et al., 2014; Besis et al., 2016; Han et al., 2018). PMF has the following advantages: each data point is given an uncertainty-weighting; the factors in PMF

are not necessarily orthogonal to each other and there is no non-negativity constraint with PMF. In the present study, PMF 5.0 (US EPA) was used to apportion the contributions of different sources to $PM_{2.5}$ in the atmosphere. The matrix X represents an ambient data set in which i represents the number of samples and j the number of chemical species. The goal of multivariate receptor modeling is to identify a number of sources (p), the species profile (f) of each source and the amount of mass (g) contributed by each source to each individual sample as well as the residuals ($e_{ij}$), as following equation:

$$X_{ij} = \sum_{k=1}^{p} g_{ik} f_{kj} + e_{ij} \qquad (S1)$$

The PMF solution minimizes the objective function Q based on these uncertainties (u):

$$Q = \sum_{i=1}^{n} \sum_{j=1}^{m} \left[ \frac{Xi_j \sum_{k=1}^{p} g_{ik} f_{kj}}{u_{ij}} \right]^2 \qquad (S2)$$

The input data files of PMF consist of concentrations and uncertainty matrices, and the uncertainty data were calculated as Equation (S3) as suggested by PMF User Guide. The missing values were represented by average values, while measurements below MDL (method detection limit) were replaced by two times of the corresponding MDL values. The "weak" variables were down-weighted, while "bad" variables were omitted form the analysis process.

$$\begin{cases} Unc_i = \dfrac{5}{6} \times MDL_i & C \le MDL_i \\ Unc_i = \sqrt{(C_i \times ErrorFraction)^2 + (MDL_i/2)_2} & C > MDL_i \end{cases} \qquad (S3)$$

The model was run 20 times with 25 random seeds to determine the stability of goodness-of-fit values. It is necessary to test different numbers of sources to find the optimal number of sources which produces the most reasonable results. If the number of sources is estimated properly, the theoretical Q value should be approximately the number of degrees of freedom or the total number of data points. Five to eleven factors were examined, and eight factors were found to be the most appropriate and most reasonably interpretable. Q (True) is the goodness-of-fit parameter calculated including all points, while Q (Robust) is the goodness-of-fit parameter calculated excluding points not fit by the model, Q (Robust) and Q (True) were 1,752.4 and 1,812.9, respectively. Additionally, approximately 98% of the residuals calculated by PMF were within the range of -3 to 3, indicating a good fit of simulated results. The factor did not show oblique edges, suggesting there were little rotation for the solution. All these features implied the model simulation result was acceptable. " in Section S1 in the revised Supplementary Materials.

**2. Page 23, line 449-457. This manuscript stated that the PMF$_P$ and PMF$_T$ profiles are similar, and should be attributed to the fact that the major NPOCs are enriched in particle phase. This might not apply for factor 6, 7 and 8 (Figure 8f, g and h). The impact of G-P partitioning process will mainly affect the factors highly loaded with low molecular weight species. So the author might need to discuss the impacts of G-P partitioning on these factors.**

*Response:* The lower molecule weight species in gas-particle partitioning were more susceptible to be influenced by ambient temperature, hence light NPOCs show large fugacity from aerosol surfaces. In the present research, factor 8 was recognized as "Light NPOCs" for the characterization of high load of light NPOCs compounds. However, due to PM$_{2.5}$ aerosols in this study was mainly conducted in the cold period of high-frequency haze episodes, the resolved factor profiles between PMF$_P$ and PMF$_T$ model were similar, even for the light NPOCs factor. The discussion of impact gas-particle partitioning on these factor can be seen detailedly in line 448-456 in the Section of "3.4.3 Assessing impacts of gas-particle partitioning on source apportionment" in this original manuscript.

However, for the factor 6 and 7, namely "Biomass burning" and "Shipping and diesel exhaust", which were characterized by inorganic salts (Cl$^-$ with K$^+$) and heavy metals (Ni with V), respectively. The tracers for these factors could not be partitioned between phases, despite several light NPOCs species took relative medium to high factor loads, their impacts caused by gas-particle partitioning should be ignored.

**3. From the title, it seems that the manuscript focused on the size distribution, G-P portioning of NPOCs, and the application of NPOCs on source apportionment of PM$_{2.5}$. While this study did not measure the gas-phase NPOCs, the gas-particle portioning is only simulated basing on Pankow's theory, and could not be validated. As such, it might not be appropriate to put G-P partitioning in the title, or we can say "G-P partitioning simulation", or "the impacts of G-P portioning on source apportionment". Size distribution was measured for NPOCs, which should be related to other parts of this manuscript. For example, does the size distribution help to explain the PMF results?**

*Response:* We still think the original "Non-polar organic compounds in aerosols in a typical city of Eastern China: Size distribution, gas-particle partitioning and tracer for PM$_{2.5}$ source apportionment" was very proper, for two major reasons. 1). The size-distribution and gas-particle partitioning of NPOCs was really two major research contents in this study. The gaseous phase of NPOCs for the corresponding 13-staged aerosols were not measured in this study, for the season of it is still almost impossible to collect different size-specific particulate and the corresponding gaseous NPOCs simultaneously. In fact, we adopted the classical gas-particle partition model to simulate the abundance of gaseous NPOCs, and explore the particle fraction (φ) of NPOCs with typical organic matter parameters in urban, rural and background areas. Just as PAHs, alkanes, hopanes and steranes could be called as "NPOCs" in this study, though not all kinds of NPOCs species

were analyzed.

2). The NPOCs were used as tracers for the source identification of PM$_{2.5}$ by specific specie ratios and receptor model (PMF), but not source apportionment of NPOCs themselves. If the title was changed to "gas-particle partitioning simulation" or "the impacts of gas-particle partitioning on source apportionment", either it lost the key section of "source identification of PM$_{2.5}$" or it is wrong for generalizing the major research objects for this study.

The size-specific distributions of NPOCs have important influence on their gas-particle partitioning and photo-degradation. Also, combining the characterized species ratios and model extractions, size-specific distributions of NPOCs have relation to aerosols source identifications. The size-distribution of NPOCs was tightly related to the parts of Sections of "Degradation of organics", "Gas-particle partitioning" and "PMF source apportionment" in this study.

**4. Diagnostic rations of n-alkanes, PAHs and hopanes were intensively used to evaluate the sources of NPOCs in previous work. The criteria of diagnostic ratios are qualitative and confusing.**

*Response:* Despite diagnostic ratio was relatively a bit empirical and rough when used for the source identification in some cases, it could easily provide useful information in most situations. Additionally, the combined use of diagnostic ratios of NPOCs and PMF model would provide mutual authentication.

**5. Besides the above comments, the lack of enough novelty is the main issue for this work. The size distribution and diagnostic ratios of NPOCs in typical Chinese cities were intensively investigated (Bi et al., 2005; Zhou et al., 2005; Wang et al., 2009a, b, 2011; Duan et al., 2012). The impacts of G-P portioning of semi-volatile organic compounds (SVOCs) on PMF source apportionment have been observed and validated by Xie et al. (2013, 2014), and the method of using gas + particle phase SVOCs have been intensively applied in PMF source apportionment studies (Gao et al., 2015; Wang et al., 2016; Zhai et al., 2016). Moreover, this work did not provide any new phenomenon or viewpoints that add our knowledge on size distribution or G-P partitioning of NPOCs, or sources apportionment using NPOCs data.**

*Response:* China is suffering severe complex atmospheric pollution, e.g. the persistent heavy haze pollution in Eastern China. Due to the heterogeneous developments of industrial, economic, geomorphic and environmental conditions, cities are facing different environmental pressures and situations. However, researches of NPOCs were currently focused on megacities in relatively developed costal areas in China, leaving most medial cities be ignored. Undoubtedly,

systematically analyzing aerosols bound NPOCs and learning their tracers for the source apportionment of $PM_{2.5}$ in a typical medium city, has great academic and practical values without doubt.

NPOCs as one important class of particles were rather chemically stable, which have been reported by numerous researchers, several researchers use them as tracers for $PM_{2.5}$ source apportionment. Just as reviewer suggested (Table R2), there were numerous researches about molecular compositions, sized distributions of organic compounds and their effects on G/P partitioning have been published. However, these researches have their unique research interests, showed different focuses to our manuscript (Table R1), e.g. Wang et al., (2011b), Wang et al., (2009) and Wang et al., (2015) analyzed the concentrations of NPOCs and researched their characterizations. Wang et al., (2011a) reported concentrations, characterization and the size specific (0.4-9.0 μm) distributions of n-alkanes, PAHs and hopanes in three different typical sites, they neither evaluated their gas-particle partitioning of these compounds, nor investigated their sources and the corresponding contribution. Xie et al., (2014) evaluated the gas-particle partitioning process of six species of PAHs, twelve species of n-alkanes, hopanes and steranes, explored the partitioning impacts on their source apportionment, and got five NPOCs species profiles (odd alkane, light SVOCs, n-alkane, PAHs and sterane).

This manuscript systematically provides new information on the profiles and characterizations of $PM_{2.5}$-associated NPOCs, evaluated their size-specific distributions and impacts on the gas-particle partitioning, found the effects of gas-particle partitioning and degradation were not apparent o the source apportionment. As best as our knowledge, this is the first research systemically analyzing the characterization, size-specific distribution, gas-particle phase partitioning of NPOCs, and exploring effects of partitioning between tracers for the aerosol source identifications. Based on this manuscript, it will help us to identify the more accurate sources of aerosols and asses the contributions from each source, provide information for further targeted optimized emission control strategies.

**Table R2**   Comparison between related studies with this manuscript

| ID | Studies | Sampling site | Sampling duration | Analysis method | PAHs | n-Alkanes | Hopanes | Steranes | Size-distribution | Gas-particle partition | Source identification | Remark |
|---|---|---|---|---|---|---|---|---|---|---|---|---|
| ① | Wang et al., 2011a; ACP | Baoji city, China | 11-14, Jan. 12-20, Feb. 12-24 Apr. 2008 | Ultrasonication Extraction+GC/MS | 16 species | C18-C36 | 5 species | / | TSP; 0.4-9.0 μm, nine stage; | / | / | |
| | | Mount Tai, China | 22-29 Jun. 2006; 12-24 Jan. 2008 | | | | | | | | | |
| | | Okinawa Island, Japan | 18 Mar. -12 Apr. 2008 | | | | | | | | | |
| ② | Wang et al., 2011b; EST | Nanjing city, China | 1-17 Jan. 2007 12-14 Oct. 2007 | Ultrasonication Extraction+GC/MS | 16 species | C18-C34 | / | / | 0.4-9.0 μm, nine stage; | / | / | |
| ③ | Wang et al., 2009; Tellus, B | The same to study ① | The same to study ① | Ultrasonication Extraction+GC/MS | / | / | / | / | 0.4-9.0 μm, nine stage; | / | / | Mainly about sugar, sugar-alcohol, carbonxylic acid |
| ④ | Wang et al., 2015; EST | Nanjing city, China | Jul. 2004 - Jan. 2005 | Solvent extraction+ GC/MS | / | C18-C36 | / | / | / | / | / | Tracers for source identification |
| ⑤ | Wang et al., 2006; EST | 14 Chinese cities | 2 days in winter + 2 days in summer, 2003 | Ultrasonication Extraction+GC/MS | 18 species | C16-C35 | C27-C32 | / | / | / | / | Other sugars and so on |
| ⑥ | Gao et al., 2015; AE | Guizhou city, China | 28 Nov. - 23 Dec. 2009 | Ultrasonication Extraction+GC/MS | 13 species | / | 4 species | / | / | / | Factor identification + correlation analysis | |
| ⑦ | Xie et al., 2014; EST | Denver, USA | Aug. 2012- Jul. 2013 | / | 6 species | 12 species | 5 species | 5 species | / | / | PMF model | Extracted five species profiles |
| | This manuscript; ACP | Jiujiang city, China | Sep. - Dec. 2016 | TD-GC/MS, without solvent extraction | 15 species | 30species, C11-C40 | 5 species | 5 species | 13 stage, 0.01-18 μm | Partitioning between all these NPOCs | Tracers + PMF model | Extracted 8 factors for $PM_{2.5}$ aerosols |

We thank Referee #1 for his good suggestions and his/her careful reading our article.

**Reference:**

① Wang, G., Kawamura, K., Xie, M., Hu, S., Gao, S., Cao, J., An, Z., Wang, Z., 2009a. Size-distributions of n-alkanes, PAHs and hopanes and their sources in the urban, mountain and marine atmospheres over East Asia. Atmospheric Chemistry and Physics J1 - ACP 9, 8869-8882.

② Wang, G., Kawamura, K., Xie, M., Hu, S., Cao, J., An, Z., Waston, J.G., Chow, J.C., 2009b. Organic Molecular Compositions and Size Distributions of Chinese Summer and Autumn Aerosols from Nanjing: Characteristic Haze Event Caused by Wheat Straw Burning. Environmental Science & Technology 43, 6493-6499.

③ Wang, G., Kawamura, K., Xie, M., Hu, S., Li, J., Zhou, B., Cao, J., An, Z., 2011. Selected water-soluble organic compounds found in size-resolved aerosols collected from urban, mountain and marine atmospheres over East Asia. Tellus B 63, 371-381.

④ Wang, G., Kawamura, K., 2005. Molecular Characteristics of Urban Organic Aerosols from Nanjing: A Case Study of A Mega-City in China. Environmental Science & Technology 39, 7430-7438.

⑤ Wang, G., Kawamura, K., Lee, S., Ho, K., Cao, J., 2006. Molecular, Seasonal, and Spatial Distributions of Organic Aerosols from Fourteen Chinese Cities. Environmental Science & Technology 40, 4619-4625.

⑥ Gao, B., Wang, X.-M., Zhao, X.-Y., Ding, X., Fu, X.-X., Zhang, Y.-L., He, Q.-F., Zhang, Z., Liu, T.-Y., Huang, Z.-Z., Chen, L.-G., Peng, Y., Guo, H., 2015. Source apportionment of atmospheric PAHs and their toxicity using PMF: Impact of gas/particle partitioning. Atmospheric Environment 103, 114-120.

⑦ Xie, M., Hannigan, M.P., Barsanti, K.C., 2014. Impact of Gas/Particle Partitioning of Semivolatile Organic Compounds on Source Apportionment with Positive Matrix Factorization. Environmental Science & Technology 48, 9053-9060.

⑧ Huang R J, Zhang Y, Bozzetti C, 2014. High secondary aerosol contribution to particulate pollution during haze events in China. Nature, 514(7521):218.

We tried our best to improve the manuscript and made some changes in the manuscript. These changes will not influence the content and framework of the paper. We appreciate for Editors/ Reviewers' warm work earnestly, and hope that the correction will meet with approval. Once again, thanks very much for your comments and suggestions.

Yours sincerely,

Best regards!

Jinping Cheng    Prof./Doctoral Supervisor

Tel: +86 21 54743936

Fax: (86 21) 5474 0825

E-mail: jpcheng@sjtu.edu.cn

Add.:800 Dongchuan Road, Minhang District Shanghai, China

   Room 508, School of Environmental Science and Engineering, Shanghai Jiao Tong University

---

## Author Response (AR2)

**A list of responses for comments from editors and reviewers**

**Dear Editors and Reviewers:**

Thank you for your letter and for the reviewers' comments concerning our manuscript entitled "***Non-polar organic compounds in aerosols in a typical city of Eastern China: Size distribution, gas-particle partitioning and tracer for PM$_{2.5}$ source apportionment***" (No. acp-2017-908). These comments are valuable and very helpful for revising and improving our paper, as well as the important guiding significance to our researches. We have studied comments carefully and have made correction, the correction in the revised manuscript was marked-up with **BLUE** color which we hope meet with approval.

The main corrections in the paper and the responds to the reviewer's comments are as flowing:

**Content**

**Twenty-seven Pages, Three parts.**

**Suggestions for revision or reasons for rejection**

**-Reviewer #1**

**This study investigated non-polar organic compounds including PAHs, alkanes, hopanes and steranes, by using TD-GC/MS technology, in different size-segregated particles from a city (Jiujiang) in eastern China. While the study did provide some information on the characteristics of organic species in a Chinese city, the manuscript in its present form, in my opinion, needs a further clarification and revision before its suitable for publication.**

   *Response:* Thanks for reviewer's suggestion, the manuscript has been revised according to editor's and reviewer's suggestions carefully, and the revised section was marked with **BLUE** color in this revised manuscript.

**Novelty of the study:**

**Without doubt that it is valuable to investigate the occurrence of tracer organic species in environment to better reflect the scientific understanding on their sources, impacts and roles in air pollution source apportionment, however, it is still a concern in my opinion that the novelty of this study is very limited. The methodologies including sampling, laboratory analysis and data interpretation are not new. Results and most conclusions are as expected and may be limited to the study area and period. So, what makes such a measurement in a local city novel and scientific that desires for the publication in ACP? [1] What's the implication of this study that is important, may be generated and promote our understanding on the characterization of tracer organics or air pollution? [2] The authors are suggested to rethink about these and add appropriate discussions in the paper before its suitable for publication.**

   *Response:* Thanks for reviewer's suggestion. (1). As best as our knowledge, this is the first research systemically analyzing size-specific distribution, photo-degradation and gas-particle partitioning of NPOCs, and evaluating their

effects on PM$_{2.5}$ source apportionment. This study was conducted in Jiujiang, a middle scale city in Eastern China, which was more typical than metropolis cities (e.g. Shanghai, Nanjing) for its more complicated emission sources and topography. The PM$_{2.5}$ sources including coal combustion, vehicle emission, oil combustion, ship emission, dust, industrial exhaust, biomass burning, secondary formation and so on. In which, some origins were banned in metropolis cities, such as biomass burning was banned in Shanghai. For the topography, recent researches found that the mountain topography enhanced the pollution degree and lengthened the pollution period, since mountain suited surrounding cities would hinder polluted air mass transportation, condensed under the atmospheric boundary layer, enhancing the atmosphere pollution. The complicated emission sources, typical topography with plain in northern direction, Poyang Lake in the east direction and Mount Lu in the southern direction, all contributed an excellent typical case for researching PM$_{2.5}$ associated NPOCs.

(2). This study found NPOCs distributed in 0.56–1.00 μm were majorly formed from condensation of combustion products. OC/EC promote NPOCs partitioned to particle phase in accumulation mode. Using the predicted gas + measured particle phases NPOCs as input data for PMF analysis, received better factor profiles than particle NPOCs only. These findings will help us accurately identify the potential sources of aerosols and then asses the contributions from each source.

According to reviewer's suggestion, the discussions were revised and further analyzed, the major modifications were as following.

[revised manuscript text omitted]

**Study site and period:**

**(3). Why the city is typical in eastern China area? Shall the situation and organic compounds characteristics differ in the studied city from those like Shanghai and Nanjing in east China?**

*Response:* As mentioned above, this study was conducted in Jiujiang, a middle scale city with 4.83 million resident population in 2015, which was the second largest city in Jiangxi Province (there are 6 provinces and one municipality in Eastern China).

For emission sources. The $PM_{2.5}$ sources including coal combustion, vehicle emission, oil combustion, ship emission, dust, industrial exhaust, biomass burning, secondary formation and so on. In which, some origins were banned in metropolis cities, such as biomass burning was banned in Shanghai. There were over 700 thousand motor vehicles in 2015, vehicle emissions were significant; also, Jiujiang Harbor is one huge harbor in the middle of Yangtze River, the ship emission also contributed to the emissions. According to local preliminary statistics, the gross industrial standard coal consumption in Jiujiang amounted to 7.80 million tons in 2015. In contrast to the coastal megalopolis cities (e.g. Shanghai, Nanjing), the domestic coal combustion and biomass combustion were not banned which would emit NPOCs into atmosphere. In Jiujiang, petrochemical industry, which can process approximately five million tons of crude oil per year, is located at the northeast part of the city and in upwind direction. Mount Lu and other surrounding forest provide abundant plant origins for NPOCs.

For the topography, recent researches found that the mountain topography enhanced the pollution degree and lengthened the pollution period, since mountain suited surrounding cities would hinder polluted air mass transportation, condensed under the atmospheric boundary layer, enhancing the atmosphere pollution. The air mass transported from Northern China which brought large amounts of NPOCs, would be accumulated in this city areas for the blocking effect caused by high mountains. The complicated emission sources, typical topography with plain in northern direction,

Poyang Lake in the east direction and Mount Lu in the southern direction.

All these mentioned contributed an excellent typical case in this city for researching PM$_{2.5}$ associated NPOCs.

**(4). The sampling was from Sep. to Dec. in 2016 (line 102-103), thus this study may be considered as a wintertime study, and results cannot represent a general situation in the city. This info. should be clarified in the title and abstract sections.**

*Response:* The sampling in this study conducted from Sep. to Dec. 2016, mainly represents autumn and early winter time in local climate. Considering reviewer's suggestion, the sampling time in the abstract was clarified, it was modified as "The samples were majorly collected in autumn and winter in a typical city of Eastern China." in line 15-16 of the revised manuscript. Also, other corresponding statement of sampling time was changed, e.g. "Compared with long time investigation, this study was mainly focused in cold season, which would lead to relative high abundance of particle NPOCs with small variation." in line 507-508 of the revised manuscript. The original title "Non-polar organic compounds in aerosols in a typical city of Eastern China: Size distribution, gas-particle partitioning and tracer for PM$_{2.5}$ source apportionment" was changed to "Non-polar organic compounds in autumn and winter aerosols in a typical city of Eastern China: Size distribution and impact of gas-particle partitioning on PM$_{2.5}$ source apportionment" in the revised manuscript.

**NPOC analysis using TD-GC/MS:**

**(5). Line 142-144, Table 2 may be moved to SI as these parameters are mot key part of the study. I'd like to suggest to a new table listing the retention time and quantification ions for each species in the paper. Though the full scan method has been adopted in the study using TD coupled with GC/MS or MSMS technologies, the**

**accuracy of the quantification is still a big problem. It would be also helpful to provide the mass spectra figure of a representative sample.**

*Response:* Thanks for reviewer's suggestion. The original Table 2 was moved to the Section S2 in the Supplementary Materials. Additionally, the retention time and quantification ions for each NPOCs species was added to this revised table (see **Table R1**). According to reviewer's suggestion, the mass spectra figures of Pyrene, Coronene, n-Docosane and n-Dotriacontane were depicted in the Section S2 in Supplementary Materials, as shown in **Fig. R1**.

**Table R1.** Abbreviation, $P^o_L$ and $\Delta H_0$, retention time and quantification ion informationinformation for individual NPOCs

| Species | Abb. | $P^o_L$ [a] | $\Delta H_0$ [b] | Base peak (m/z) | Retention Time (RT) |
|---|---|---|---|---|---|
| **PAHs** | | | | | |
| Fluorene | FLO | 1.10E–01 | 84.9 | 166 | 19.25 |
| Phenanthrene | PHE | 2.57E–02 | 88.9 | 178 | 24.41 |
| Anthracene | ANT | 1.21E–03 | 99.7 | 178 | 24.56 |
| Fluoranthene | FLU | 1.60E–03 | 98.3 | 166 | 28.03 |
| Pyrene | PYR | 7.60E–04 | 97.9 | 202 | 28.66 |
| Benz[a]anthracene | BaA | 3.45E–05 | 108 | 228 | 32.30 |
| Chrysene | CHR | 1.36E–06 | 118.8 | 228 | 32.41 |
| Benzo[b]fluoranthene | BbF | 1.00E–06 | 119.2 | 252 | 35.32 |
| Benzo[j+k]fluoranthene | BkF | 4.66E–06 | 113 | 252 | 35.37 |
| Benzo[a]fluoranthene | BaF | 4.66E–05 | 113 | | 36.13 |
| Benzo[e]pyrene | BeP | 7.89E–07 | 117.9 | 252 | 36.01 |
| Indeno[1,2,3-cd]pyrene | IcdP | 1.42E–06 | 124 | 276 | 39.22 |
| Dibenz[a,h]+[a,c]anthracene | DahA | 4.93E–09 | 134.1 | 278 | 39.00 |
| Benzo[ghi]perylene | BghiP | 1.01E–08 | 129.9 | 276 | 38.91 |
| Coronene | COR | 3.56E–10 | 143.2 | 300 | 28.71 |
| **iso–Alkane** | | | | | |
| Pristane | $C_{19}H_{40}$ | / [c] | / | 57 | 23.24 |
| Phytane | $C_{20}H_{42}$ | / | / | 57 | 24.69 |
| **Hopane** | | | | | |
| αβ–Nnorhopane | $C_{29}$–αβ–NOR–H | 2.74E–06 | 126 | 191 | 37.77 |
| αβ–Hopane | $C_{30}$–αβ–H | 1.01E–06 | 130 | 191 | 38.54 |
| αβ–22R–Homohopane | $C_{31}$—αβ–R | 3.85E–07 | 134 | 191 | 39.56 |
| ab 22S–Homohopane | $C_{31}$—αβ–S | 3.85E–07 | 134 | 191 | 39.70 |
| 22,29,30–Trisnorhopane | Tm | 1.93E–05 | 117 | 191 | 36.63 |
| **Sterane** | | | | | |

| | | | | | |
|---|---|---|---|---|---|
| ααα–20R Cholestane | ααα–20R–C | 2.03E–05 | 121 | 217 | 37.29 |
| αββ–20R Cholestane | αββ–20R–C | / | / | 218 | 37.66 |
| αββ–20R24S–Methylcholestane | αββ–20R–MEC | 7.60E–06 | 125 | 218 | 36.58 |
| aaa 20R24R–Ethylcholestane | ααα–20R–EC | / | / | 217 | 37.29 |
| αββ–20R24R–Ethylcholestane | αββ–20R–EC | 2.84E–06 | 130 | 218 | 37.66 |
| **n–Alkanes** | | | | | |
| n–Undecane | $C_{11}H_{24}$ | / | / | 57 | 12.39 |
| n–Dodecane | $C_{12}H_{26}$ | / | / | 57 | 13.92 |
| n–Tridecane | $C_{13}H_{28}$ | / | / | 57 | 16.10 |
| n–Tetradecane | $C_{14}H_{30}$ | / | / | 57 | 18.15 |
| n–Pentadecane | $C_{15}H_{32}$ | / | / | 57 | 20.26 |
| n–Hexadecane | $C_{16}H_{34}$ | / | / | 57 | 21.63 |
| n–Heptadecane | $C_{17}H_{36}$ | / | / | 57 | 23.15 |
| n–Octadecane | $C_{18}H_{38}$ | / | / | 57 | 24.55 |
| n–Nonadecane | $C_{19}H_{40}$ | / | / | 57 | 25.87 |
| n–Eicosane | $C_{20}H_{42}$ | / | / | 57 | 27.12 |
| n–Heneicosane | $C_{21}H_{44}$ | / | / | 57 | 28.33 |
| n–Docosane | $C_{22}H_{46}$ | 3.24E–03 | 115 | 57 | 29.43 |
| n–Tricosane | $C_{23}H_{48}$ | 1.22E–03 | 120 | 57 | 30.61 |
| n–Tetracosane | $C_{24}H_{50}$ | 4.66E–04 | 124 | 57 | 31.55 |
| n–Pentacosane | $C_{25}H_{52}$ | 1.72E–04 | 129 | 57 | 32.43 |
| n–Hexacosane | $C_{26}H_{54}$ | 6.59E–05 | 133 | 57 | 33.09 |
| n–Heptacosane | $C_{27}H_{56}$ | 2.53E–05 | 137 | 57 | 33.36 |
| n–Octacosane | $C_{28}H_{58}$ | 9.42E–06 | 142 | 57 | 33.50 |
| n–Nonacosane | $C_{29}H_{60}$ | 3.55E–06 | 146 | 57 | 35.47 |
| n–Triacontane | $C_{30}H_{62}$ | 1.32E–06 | 151 | 57 | 37.31 |
| n–Hentriacontane | $C_{31}H_{64}$ | 4.96E–07 | 155 | 57 | 39.24 |
| n–Dotriacontane | $C_{32}H_{66}$ | 1.93E–07 | 160 | 57 | 37.66 |
| n–Tritriacontane | $C_{33}H_{68}$ | 7.09E–08 | 164 | 57 | 40.22 |
| n–Tetratriacontane | $C_{34}H_{70}$ | 2.63E–08 | 169 | 57 | 38.75 |
| n–Pentatriacontane | $C_{35}H_{72}$ | 1.00E–08 | 173 | 57 | 40.21 |
| n–Hexatriacontane | $C_{36}H_{74}$ | 3.75E–09 | 177 | 57 | 41.33 |
| n–Hepatriacontane | $C_{37}H_{76}$ | 1.42E–09 | 182 | 57 | 42.82 |
| n–Octatriacontane | $C_{38}H_{78}$ | 5.37E–10 | 186 | 57 | 43.55 |
| n–Nonatriacontane | $C_{39}H_{80}$ | 2.03E–10 | 191 | 57 | 45.13 |
| n–Tetracontane | $C_{40}H_{82}$ | 7.60E–11 | 195 | 57 | 46.21 |

[a]: pure compound vapor pressure, unit of Pa at 298 K, cited from And and Hanshaw, 2004, Xie et al., 2013;

[b]: vaporization enthalpy, unit of (KJ mol$^{-1}$) at 298 K, cited from Xie et al., 2013, Wang et al., 2016;

[c]: "/" means lack of related data.

[Figure]

[Figure]

**Fig. R1.** The mass spectra of selected NPOCs (a for Pyrene, b for Coronene, c for n‑Docosane, d for n‑Dotriacontane)

**(6). Line 178-is the spiked solution from the standard chemicals or the SRM (line 166-SRM particles)?**

**QA/QC-are there any blank filters from the field sampling?**

*Response:* The statement of spiked solution in original line 178 referred to the Standard Reference Materials (SRM),

including SRM 2260A, SRM 1494 and SRM 2266. Considering reviewer's suggestion, this depiction was changed to

"The analytical recovery was calculated via spiking a known amount of the SRM solution to blank filter" in line

176-177 of the revised manuscript.

In the Quality assurance and quality control section, not only blank filters from filed sampling, but also transport blank and laboratory blank filters were analyzed in this study. It was depicted as "Field blanks were collected by keeping blank filters in the sampler for the same duration at sampling site. Additionally, both transport and laboratory blank filters were analyzed, and all the data reported in this study were corrected according to the results." in line 159-162 in the revised manuscript.

**Gas-particle partitioning**

**(7). Without a direct measurement of gaseous organics, the whole discussion and interpretation on the gas-particle partitioning in this paper is very weak and inconclusive. There are also many other gas-particle partitioning models, of which there are very rough estimated numbers for some key parameter. The authors may still keep a small part of discussion on the partitioning and its impacts on the PMF results, however, it is not suggested to highlight this in the title and a long paragraph in the results/discussion part. Those only increased the length of the paper but not the depth and significance of the study, and to some extent have a negative impact of the manuscript organization.**

*Response:* Due to the high subcooled liquid vapor pressure or octanol-air partitioning coefficient value, heavy NPOCs, e.g. long chain n-alkane (>27 C), PAHs (molecular weight >252), hopane and sterane are mostly in particle phase. There was limited study about gaseous NPOCs have been reported. When the measured gaseous phase NPOCs data were not available, the predicted gaseous NPOCs from gas-particle model provides as a good substitution.

Among different gas-particle partitioning models, the one developed by Pankow (1994 a,b) was the most widely adopted for the partitioning of semi-volatile organic compounds (SVOCs), e.g. NPOCs, PCBs, PBDEs and divalent mercury. This gas-particle partitioning model took account of major affecting factors, such as molecular weight of the

absorbing OM phase, the mole fraction scale activity coefficient (ζ) of each compound in the absorbing OM phase, vapor pressure of each pure compound (P$^o$L), ambient temperature (T) and ideal gas constant (R).

In this study, only particle associated NPOCs were measured, without measured gas data to explore gas-particle partitioning process would inevitably bring some uncertainties. However, we do not think the whole discussion was inconclusive, since numerous researches suggested that this partitioning model predicted similar gaseous values comparable to the measured ones. For example, Xie et al. (2014) found the factor contributions were consistent from both the "measured total" and "predicted total" SVOCs, both of which reduced the influence from gas-particle partitioning.

Considering reviewer's suggestion, the discussion on the partitioning of NPOCs was shortened, and it was not highlighted in the abstract and conclusions. For example, the statement of "Gas-particle partitioning model showed that the particle-phase fraction (φ) of light molecular weight NPOCs ranged from 2.4% to 62.5%, while that of heavy NPOCs accounted for more than 90.0%." in the original abstract, "Gas-particle partitioning model showed that the calculated particle-phase fraction (φ) of the light molecular weight NPOCs ranged between 2.4% to 62.5%, while those of heavy PAHs, long chain n-alkanes, hopanes and steranes were high (>90.0%)." in the original conclusions, were deleted in the revised manuscript. The words of section "3.4.1 Gas-particle partitioning" (387 words) was shortened to ~a half of the original one, and the figure 7 was also depleted in the revised manuscript.

**Source apportionment.**

**(8). The limitations in those apportionment methods including ratios as well as PMF have been widely recognized, however, these are still widely used nowadays. The limitations are briefly mentioned in some parts (lines 278-280). It is necessary and strongly suggested to have a separate section discussing the limitation of the study including**

**sampling periods and methods used in source apportionment discussion.**

*Response:* According to reviewer's suggestion, the limitations of PMF model, sampling time and gas-particle partitioning effect were added, "3.5 Limitation and implication" in lines 494-511 in the revised manuscript. It was depicted as "In this work, we confirmed that using the total (gas+ particle) NPOCs as input data for receptor model provides a better source apportionment than using only particle phase. However, the predicted gas NPOCs from gas-particle partitioning may bring some uncertainty. For example, the partitioning process are strongly affected by the particle properties, e.g. particle size, organic carbon compounds and the prevailing ambient temperature. For size-distribution, the PM$_{2.5}$ associated NPOCs in the 0.56–1.00 μm fraction were the most abundant, our recent study also found OC was primarily distributed in this fraction (Han et al., 2018). Abundant OC would adsorb/absorb large amounts of NPOCs, resulting particle bound NPOCs concentration increasing in this particle size.

Low temperature promotes NPOCs adsorbing/absorbing onto aerosols, while photochemical degradation of NPOCs is relatively weak in cold season. Moreover, photochemical reactions would reduce the abundances of organic marker depend on species, significantly altering the relative contribution of different sources extracted by liner source inversion. Compared with long time investigation, this study was mainly focused in cold season, which would lead to relative high abundance of particle NPOCs with small variation.

For PMF model, it has limit that could not identify potential source without preexisting tracer. Also, the relative small number of measurements might lead to some uncertainty in source apportionment. In the future, more source tracers data need to be included for the calculation of potential contributions."

**Specific ones:**

**(9). Title- as mentioned "gas-particle partitioning" is not suggested to be highlighted here. Also "tracers for PM$_{2.5}$**

**source apportionment" is a little confusing to me. The study did not investigate or evaluate the use of specific trace organics in PM$_{2.5}$ source apportionment. What they did is the use of these tracers, together with ions, to run PMF and to identify potential sources of PM$_{2.5}$. Strongly suggest the authors to rethink the title for this manuscript.**

*Response:* Considering reviewer's suggestion, the original title of "Non-polar organic compounds in aerosols in a typical city of Eastern China: Size distribution, gas-particle partitioning and tracer for PM$_{2.5}$ source apportionment", was revised as "Non-polar organic compounds in autumn and winter aerosols in a typical city of Eastern China: Size distribution and impact of gas-particle partitioning on PM$_{2.5}$ source apportionment" in the revised manuscript.

**(10). Line 18-19, suggest to add the quantitative percentages, and the implication of different size distributions between PAHs and hopanes, - sources?**

*Response:* Thanks for reviewer's suggestion, the original description of " the middle chain length n–alkanes (C$_{25}$–C$_{34}$) were the most abundant in n-alkanes. PAHs and n-alkanes were majorly distributed in 0.56–1.00 μm fraction. ∑(hopanes+steranes) were associated with the 0.32–1.00 μm fraction." was changed to "the middle chain length n–alkanes (C$_{25}$–C$_{34}$) were the most abundant (72.3%) in n-alkanes. PAHs and n-alkanes were majorly distributed in 0.56–1.00 μm fraction, while ∑(hopanes+steranes) were associated with the 0.32–1.00 μm fraction, suggesting condensation of combustion products was their important origins." in line 18-20 in the revised manuscript.

**(11). Lines 23-24, no new information added to the current knowledge as this trend has been widely recognized by the researches in this area. Also high uncertainties in these quantitative numbers as this is from a simple model estimation. Suggest to remove.**

*Response:* According to reviewer's suggestion, the statement of "Gas-particle partitioning model showed that the particle-phase fraction (φ) of light molecular weight NPOCs ranged from 2.4% to 62.5%, while that of heavy NPOCs

accounted for more than 90.0%. " was deleted in the revised manuscript.

**(12). Line 109- "Han et al., 2018" is missing in the reference section**

*Response:* Thanks for reviewer's suggestion, the reference "Han, D., Zhang, J., Hu, Z., Ma, Y., Duan, Y., Han, Y., and Cheng J.: Particulate mercury in ambient air in Shanghai, China: Size-specific distribution, gas-particle partitioning, and association with carbonaceous composition. Environ. Pollut., 238, 543-553, 2018." was added in lines 547-549 in the Section of Reference of the revised manuscript.

**(13). Lines 231-232, provide references**

*Response:* The references were added in lines 231-232 of the revised manuscript, revised as "In this study, c = 17.2 Pa cm$^{-1}$, and $\theta$ is $1.1\times10^{-5}$, $1.5\times10^{-6}$ and $4.2\times10^{-7}$ (cm$^2$ cm$^{-3}$) for urban area, rural area and background, respectively (And and Hanshaw, 2004; Xie et al., 2013)."

**(14). Line 236, is this a daily average or seasonal?**

*Response:* The average value of PM$_{2.5}$ was daily average concentration, it was revised as "The daily average PM$_{2.5}$ concentration in all sampling sites " in line 236 in the revised manuscript.

**(15). Line 237-238, the sentence "which was estimated…" may be moved to line 237 after the phase "organic matter (OM)"**

*Response:* Thanks for reviewer's suggestion, this statement was revised as "Organic matter (OM) was estimated to be 1.4 times of OC concentration (Feng et al., 2006;Huang et al., 2014), which was the most abundant component in PM$_{2.5}$, accounting for 18.8–27.8% of the total mass in this study. " in lines 237-239 in the revised manuscript.

**(16). Line 242, "daily average concentration'**

*Response:* This statement of "The total concentrations ranged from" changed to "Their daily average concentrations ranged from " in line 242 in the revised manuscript.

**(17). Lines 243-244, what're the quantitative percentages of these studies? It is interesting but also a little strange that the results were comparable in a fast-developed area (PRD), the marine aerosol (south China sea study), and the present city in inland. Please provide results and study periods in these literature and discuss if applicable.**

*Response:* The quantitative percentages of NPOCs to OM in PRD South China Sea were 0.1–4.2% and 0.8–1.7%, respectively. Considering reviewer's suggestion, this description was revised as "This was consistent with the measurement results (0.1–4.2%) of NPOCs in Pearl River Delta (PRD, China) (Wang et al., 2016), over a two-year period from 2011 to 2012. Similarly, Zhao et al., (2016) found that the NPOCs varied from 19.8 to 288.2 ng m$^{-3}$, accounted for 0.8–1.7% of OM in South China Sea from Sep. to Oct. 2013." in lines 243-246 in the revised manuscript.

**(18). Lines 243-251, when comparing to literature data, it is important and necessary to clarify the study period and year of those in literature. Some data in Table 3 are the annual average, while some are the monthly or maybe daily average. A simple comparison, to the wintertime study in this paper, and the high/low contamination conclusion here is not acceptable.**

*Response:* According to reviewer's suggestion, the statement in lines 248-251 of the original manuscript, was modified by adding detailed information of each research in lines 250-260 in the revised manuscript. The revised description was "When comparing with other NPOCs measurements in China, Li et al. (2013) reported a comparable level that the daily concentration of n-alkanes, PAHs and hopanes were 97.9, 13.5 and 21.5 ng m$^{-3}$ in Hong Kong in

winter, respectively. Xu et al. (2013) measured mean n-alkanes, PAHs concentrations of 48.1±20.8, 19.0±17.5 ng m$^{-3}$ in urban Guangzhou during 16$^{th}$ Asian Games, while Feng et al. (2006) found that the daily concentrations of n-alkanes, PAHs ranged of 32.9–342.9 and 7.8–151.1 ng m$^{-3}$ in urban Shanghai in 2002–2003, respectively. Thus, this study finds generally higher PM$_{2.5}$ associated NPOCs concentrations measured in Jiujiang compared to other measurements, which may be due to this study was mainly conducted in cold season when severe atmospheric pollution episodes (including haze) frequently occurred. In addition, aerosols and NPOCs would transported to Jiujiang from Northern China where has significant amounts of coal burned and industries, through long range transport of air mass (Han et al., 2018). However, the annual average concentrations of PM$_{10}$ bound n-alkanes and PAHs in Delhi, India (Yadav et al., 2013) were 4.0 and 8.2 times higher than that in Jiujiang, respectively. "

**(19). Figure 2- is the left circle PM$_{2.5}$- inconsistent with the legend. Only 5 site results? The area of those might be adjusted to reflect the mass concentration. Also, please pay attention to the significance figures. Did the measurement technology provide an accuracy of 0.01 μg/m3?**

*Response:* Thanks for reviewer's suggestion, this original figure (**Fig. R2**) was modified by rearranging the legends to match PM$_{2.5}$ and NPOCs constitutes, changing the accuracy of the measured abundance. The modified figure was depicted as **Fig. R3**.

The PM$_{2.5}$ associated NPOCs were measured at five sites synchronously, including XY, SL, JJ, SH and WQ sites. However, the 13-staged size-specific aerosol bound NPOCs were measured at EM site.

[Figure]

**Fig. R2.** Spatial distributions of NPOCs and PM_{2.5} in Jiujiang city

[Figure]

**Fig. R3.** The percentiles of NPOCs and PM_{2.5} constituents in five sampling sites in Jiujiang city

**(20). Line 307- "emissions were similar", or the alkanes are more likely homogeneously distributed across**

**different sites?**

*Response:* According to reviewer's suggestion, this description was changed to "This small fluctuations may suggest that n-alkanes emissions were similar across different sampling sites, displayed a relative homogeneous distribution in Jiujiang." in lines 316-318 in the revised manuscript.

**(21). Line 326, the peak in "9.9-18 μm" is very tiny**

*Response:* The peak value of PAHs in 9.9-18.0 μm was much lower than that in 0.56-1.0 μm, but the peak values of this coarse mode were enhanced for n-alkanes, hopans & steranes, as depicted in Fig 4 in the revised manuscript.

**(22). Figure 5- suggest to find another way to improve the presentation of these data. for example, maybe monitoring data from the present study can be in the same white or black sign as spatial distribution is not discussed here.**

*Response:* The original figure (marked as **Fig. R4**) was revised by changing the monitor data to black sign in the revised manuscript, according reviewer's suggestion, and the revised figure was depicted in **Fig. R5**.

[Figure]

**Fig. R4.** Ratio–ratio plots of two pairs of characterized species (IcdP/BghiP and C29–αβ–NOR–H/C30–αβ–H) normalized by EC and published source profiles. **(**Tunnel I: Yu et al., 2011; Tunnel II: He et al., 2009; Residental coal: Zhang et al., 2008; Industrial coal: Zhang et al., 2008; Diesel vehicle: Fraser et al., 2002; Gasoline vehicle: Fraser et al., 2002**)**

[Figure]

**Fig. R5.** Ratio–ratio plots of two pairs of characterized species (IcdP/BghiP and C29–αβ–NOR–H/C30–αβ–H) normalized by EC and published source profiles. **(**Tunnel I: Yu et al., 2011; Tunnel II: He et al., 2009; Residental coal: Zhang et al., 2008; Industrial coal: Zhang et al., 2008; Diesel vehicle: Fraser et al., 2002; Gasoline vehicle: Fraser et al., 2002**)**

**Special thanks to you for your careful reading and good comments!**

**-Reviewer 2**

**This study measured the concentrations of non-polar organic compounds (NPOCs) in PM$_{2.5}$ and size resolved particles. Gas/particle portioning of the target compounds were also analyzed and considered during the source apportionment of PM$_{2.5}$. In general, this work is well organized and written. I would recommend this work to be accepted after minor revision.**

*Response:* Thanks for reviewer's suggestion, the manuscript has been revised according to reviewer's suggestion carefully, and the revised section was marked with **BLUE** color in this revised manuscript.

**Minor revision.**

**(1). Page 6. Table 2**

**I would suggest the author to leave this table in supporting information.**

*Response:* According to reviewer's suggestion, this table was moved to Section S2 of the Supplementary Materials.

**(2). The method for source apportionment was provided in supporting information, would the author please mention it somewhere in the manuscript?**

*Response:* Considering reviewer's suggestion, the PMF model was mentioned in the revised manuscript, depicted as "To explore the impact of gas-particle partitioning on PM$_{2.5}$ source apportionment, both single particle phase and the total (gas+ particle) NPOCs were incorporated with elemental species, inorganic ions and OC/EC, used as input data for receptor model PMF (detailed description about PMF model could be seen in Section 2 of Supplementary Materials)." in lines 423-426 of the revised manuscript.

**(3). Page 20, lines 414-420, and page 22 figure 8**

**Where are the elements and ions data from? If they are reported for the first time in this work, please provide the measurement method and QA/QC procedure. If not, please provide some references.**

*Response:* The measurement methods of OC/EC, elements and ions data were described in the section of "2.3 Determination of OC/EC and other constituents". According to reviewer's suggestion, the Quality Assurance and Quality Control was added in lines 146-156 in the revised manuscript.

The revised statement of their measurement was "OC and EC were analyzed (a round punch of 0.538 cm$^2$) using the thermal–optical– transmittance (TOT) method (NIOSH protocol, Desert Research Institute, USA) (Han et al., 2018). The instrument included a temperature- and atmosphere-controlled oven and a laser of 680 nm wavelength to generate an operational EC/OC split. The instrument was heated stepwise from start to 250 ℃ (60 s), 500 ℃ (60 s), 650 ℃ (60 s) and finally 850 ℃ (90 s) in the helium atmosphere for OC volatilization, and from start to 550 ℃ (45 s), 650 ℃ (60 s), 750 ℃ (60 s) and finally 850 ℃ (80 s) in the helium atmosphere containing 2% oxygen for EC oxidation.

Elemental compositions, including Na, K, Ca, Mg, P, Fe, Ti, Al, Pb, Cu and Zn, were determined by energy dispersive X-ray fluorescence (ED-XRF) spectrometry (Epsilon 5, Netherlands). Water soluble inorganic ions, including cations (Na$^+$, K$^+$, Mg$^{2+}$, Ca$^{2+}$, NH$_4^+$) and anions (Cl$^-$, SO$_4^{2-}$ and NO$_3^-$, NO$_2^-$), were detected by ion chromatography (IC, ISC-90, Dionex, USA). The detailed experimental procedure of OC/EC, elemental composition, inorganic ions analysis could be found in Li et al. (2017) and Han et al. (2018)."

**(4). Page 23, lines 484-486.**

**Please add some references for this statement.**

*Response:* Considering reviewer's suggestion, the references were added in lines 502-504 in the revised manuscript, the revised statement was "While gas phase oxidation reaction is much faster than heterogeneous reactions in aerosol

surface, since the uptake of heterogeneous oxidant is diffusion-limited (Robinson et al., 2006;May et al., 2012)."

**Special thanks to you for your careful reading and good comments!**

**-Other major modifications**

The total words of the whole original manuscript was 10228, while the revised manuscript was 10155 words totally.

**Abstract.**

The abstract was shortened from 279 words to 256 words, and the statement of "This study provides new information on the profiles of $PM_{2.5}$-associated NPOCs, size-specific distributions, photo-degradation and their gas-particle partitioning. This will help us accurately identify the potential sources of aerosols and then asses the contributions from each source.", was revised to "These findings highlight the emissions from different aerosols associated NPOCs origins, caused different size-specific distributions, photo-degradation and gas-particle partitioning, which further affect $PM_{2.5}$ source apportionment. Considering these effects on organic tracers will help us accurately identify the potential sources of aerosols and then asses the contributions from each source." in lines 25-28 in the revised manuscript.

**Abbreviations of the PAHs, hopanes and steranes compounds.**

The abbreviations of PAHs, hopanes and steranes in the main manuscript were given in full expressions, e.g. "(6) MDRs for PAHs source apportionment include ANT/PHE atio, PYR/FLU ratio, IcdP/BghiP ratios" was revised to "(6) MDRs for PAHs source apportionment include ANT/PHE (Anthracene/Phenanthrene) ratio, PYR/FLU (Pyrene/Fluoranthene) ratio, IcdP/BghiP (Indeno[1,2,3-cd]pyrene/Benzo[ghi]perylene) ratios" in lines 202-203 in the revised manuscript.

"BbF (5.7 ng m$^{-3}$) was the most abundant PAH species, followed by BaA (5.6 ng m$^{-3}$) and BaP (4.2 ng m$^{-3}$)" was changed as "BbF (Benzo[b]fluoranthene, 5.7 ng m$^{-3}$) was the most abundant PAH species, followed by BaA (Benz[a]anthracene, 5.6 ng m$^{-3}$) and BaP (Benzo[e]pyrene, 4.2 ng m$^{-3}$)" in lines 271-272 in the revised manuscript.

"The predominant hopane analogs were $C_{30}$–αβ–H, $C_{31}$–αβ– and $C_{29}$–αβ– NOR–H, with concentrations of 1.2±1.3,

0.8±0.7 and 0.8±0.9 ng m$^{-3}$, respectively." was modified as "The predominant hopane analogs were C$_{30}$–αβ–H

(αβ–Hopane), C$_{31}$–αβ–S (ab 22S–Homohopane) and C$_{29}$–αβ– NOR–H (αβ–Nnorhopane), with concentrations of 1.2±1.3,

0.8±0.7 and 0.8±0.9 ng m$^{-3}$, respectively."    in lines 325-326 in the revised manuscript.

We tried our best to improve the manuscript and made some changes in the manuscript. These changes will not influence the content and framework of the paper. We appreciate for Editors/ Reviewers' warm work earnestly, and hope that the correction will meet with approval. Once again, thanks very much for your comments and suggestions.

Yours sincerely,

Best regards!

Jinping Cheng    Prof./Doctoral Supervisor

Tel: +86 21 54743936                Fax: (86 21) 5474 0825

Mail: jpcheng@sjtu.edu.cn                Add.:800 Dongchuan Road, Minhang District Shanghai, China